# A mechanism of origin licensing control through autoinhibition of *S. cerevisiae* ORC·DNA·Cdc6

Jan Marten Schmidt[1,2,4,5], Ran Yang[3,5], Ashish Kumar[3], Olivia Hunker[3], Jan Seebacher [1] & Franziska Bleichert [3✉]

The coordinated action of multiple replicative helicase loading factors is needed for the licensing of replication origins prior to DNA replication. Binding of the Origin Recognition Complex (ORC) to DNA initiates the ATP-dependent recruitment of Cdc6, Cdt1 and Mcm2-7 loading, but the structural details for timely ATPase site regulation and for how loading can be impeded by inhibitory signals, such as cyclin-dependent kinase phosphorylation, are unknown. Using cryo-electron microscopy, we have determined several structures of *S. cerevisiae* ORC·DNA·Cdc6 intermediates at 2.5–2.7 Å resolution. These structures reveal distinct ring conformations of the initiator·co-loader assembly and inactive ATPase site configurations for ORC and Cdc6. The Orc6 N-terminal domain laterally engages the ORC·Cdc6 ring in a manner that is incompatible with productive Mcm2-7 docking, while deletion of this Orc6 region alleviates the CDK-mediated inhibition of Mcm7 recruitment. Our findings support a model in which Orc6 promotes the assembly of an autoinhibited ORC·DNA·Cdc6 intermediate to block origin licensing in response to CDK phosphorylation and to avert DNA re-replication.

---

[1] Friedrich Miescher Institute for Biomedical Research, Basel 4058, Switzerland. [2] University of Basel, Basel 4051, Switzerland. [3] Department of Molecular Biophysics and Biochemistry, Yale University, New Haven, CT 06520, USA. [4] Present address: Novartis Institutes for Biomedical Research, Basel 4033, Switzerland. [5] These authors contributed equally: Jan Marten Schmidt, Ran Yang. ✉email: franziska.bleichert@yale.edu

The replication of DNA in eukaryotic cells requires the bidirectional loading of replicative helicases onto origins of replication and their timely activation for replisome assembly (reviewed in refs. [1–3]). The hexameric initiator ORC first binds and bends DNA, and then associates with the co-loader Cdc6 to form a ring-shaped, heptameric initiator·co-loader complex that is competent for recruiting the Mcm2-7 helicase motor and Cdt1 to origins (reviewed in refs. [4,5]). Docking of hexameric Mcm2-7 onto ORC·Cdc6 guides DNA insertion into the central helicase pore through an opening in the Mcm2-7 ring (the Mcm2/5 gate)[6–10]. Stable association of Mcm2-7 with DNA requires the coordinated loading of two hexamers, oriented in a head-to-head fashion, to produce double hexamers that topologically entrap duplex DNA[11–13]. Origins licensed in this manner during the G1 phase of the cell cycle are primed for Dbf4-dependent kinase (DDK) and cyclin-dependent kinase (CDK) activation in S phase[14].

Apart from Cdt1 and Orc6 (a subunit of ORC), all loading factors belong to the superfamily of ATPases Associated with various cellular Activities (AAA+)[15,16]. ATP binding and hydrolysis are essential for Mcm2-7 recruitment and loading, driving the assembly of various initiation intermediates and the conformational changes needed for helicase deposition[1,5,17]. The AAA+ folds are appended by winged-helix (WH) domains in ORC, Cdc6, and some Mcm2-7 subunits that help stabilize interactions within and between different loading factors[8,18]. While all loading factors can be recruited in the absence of ATP hydrolysis to form an ORC·Cdc6·Cdt1·Mcm2-7 (OCCM) intermediate in which DNA is successfully inserted into the pore of the first Mcm2-7 ring[12,13,19,20], conversion of ATP to ADP by Mcm2-7 is critical for helicase ring closure and double hexamer formation[21–23]. By contrast, ATP hydrolysis by ORC promotes repeated Mcm2-7 loading, while that of Cdc6 mediates disassembly of unproductive loading intermediates, but neither are necessary for double hexamer assembly in vitro[21,22,24–26]. Nonetheless, ATP constitutes a critical co-factor for ORC and Cdc6, stabilizing key interactions within the ternary ORC·DNA·Cdc6 complex[27–31]. Of particular relevance are the active ATPase centers between Orc1 and Orc4, and between Cdc6 and Orc1, which are the only ATP sites that contain all the signature sequence motifs required for catalysis (Walker A and B, sensor 1 and 2, arginine finger)[16,18]. Prior structural studies have found that these ATPase sites exist in closed conformations "primed" for hydrolysis[6,8,9,32–35]; yet it is important to suppress ATPase activity sufficiently to avoid prematurely destabilizing initiation intermediates, which would lead to abortive loading events and decreased loading efficiency. The physical basis for how ATPase activity by ORC·Cdc6 is kept low to prevent premature catalysis is poorly understood.

Helicase loading (origin licensing) and activation (origin firing) are temporally separated into different stages of the cell cycle to prevent re-replication of DNA[36]. Eukaryotes have evolved various strategies to restrict Mcm2-7 loading to late M and G1 phases[37,38]. In S. cerevisiae, CDK-mediated phosphorylation of ORC aids in preventing helicase loading outside of G1 by activating a quality control pathway that also helps release unproductive loading intermediates during origin licensing in G1 by stimulating the ATPase activity of ORC·Cdc6[24,39–41]. However, whether and how posttranslational modifications alter the structure of the initiator·co-loader to prevent loading is not well characterized. Likewise, it is unknown whether defined intermediates exist that could specifically stall Mcm2-7 recruitment or deposition.

Here, we report cryo-EM structures of ARS1-bound S. cerevisiae (Sc) ORC and of the ternary complex with Cdc6 at sub-Å resolutions. Overall, the organization of ScORC·DNA and ScORC·DNA·Cdc6 resembles previous lower resolution structures[6,8,9,33,34], but ScORC·DNA·Cdc6 is seen in two distinct ring conformations that modulate DNA contacts of the co-loader Cdc6. The Orc1·Orc4 and Cdc6·Orc1 ATPase sites reside in a pre-hydrolysis state, with water molecules in non-lytic positions and a disengaged sensor 1 at the Orc1·Orc4 site, and need to be further activated for catalysis to occur. Strikingly, the N-terminal domain of Orc6 docks onto the ORC·Cdc6 ring by binding to Orc1 and Cdc6 and is situated in a manner that sterically blocks Mcm7 attachment upon CDK-mediated ORC phosphorylation. Together, our findings reveal insights into ATPase site regulation of the initiator·co-loader and provide models for how an autoinhibited ScORC·DNA·Cdc6 state helps control timely Mcm2-7 loading.

## Results and discussion

To determine a sub-3 Å structure of the eukaryotic initiator·co-loader assembly, we chose to study S. cerevisiae (Sc) ORC·DNA·Cdc6 because, in contrast to metazoan ORC, the yeast complex binds DNA in a defined sequence register[6,33,42–44], which we predicted would reduce heterogeneity and improve the attainable resolution. We reconstituted the ternary complex in vitro on an 84 bp ARS1 DNA fragment containing the ACS and B1 elements that are required for origin function[45], followed by size exclusion chromatography to enrich for DNA-bound ORC·Cdc6 assemblies (Supplementary Fig. 1). Cryo-EM analysis of the sample revealed the presence of Cdc6 in a subset of the particles; however, the weaker signal in this region of a consensus cryo-EM map (at an overall resolution of 2.3 Å) indicated compositional and/or conformational heterogeneity (Supplementary Fig. 2). Using a masked 3D classification approach of the Cdc6 region, we obtained structures of ScORC·DNA (OD) and ScORC·DNA·Cdc6 (ODC) assemblies at 2.5–2.7 Å resolution (Supplementary Figs. 2 and 3). The higher resolution of our structures compared to previous ones and the increased occupancy of the co-loader allowed de novo model building of Cdc6, correction of register shifts, and the placement of ordered solvent molecules (Supplementary Fig. 4 and Supplementary Table 1).

Our ScORC·DNA complex structure is similar to the previous 3.0 Å model with few exceptions[6]: first, we observe only weak density for the winged-helix (WH) domain of Orc2, which is sandwiched between Orc1 and Orc2 in the earlier structure of the DNA-bound yeast initiator obtained from crosslinked samples[6] (Fig. 1a). This region is more flexible in the absence of crosslinker and may thus not stably occupy the region between Orc1 and Orc2. Second, Arg 254 in the basic patch of Orc2 makes a base-specific contact with a guanine in the major groove of the B1 element of ARS1; this side chain is flipped away in ARS305-bound ORC, which contains an adenine instead of a guanine at this position[6]. Local differences in base recognition may thus contribute to observed variations of ScORC's affinity for different ARS elements[46].

**The ORC·Cdc6 ring adopts distinct conformational states.** In our cryo-EM structure of the ternary complex, Cdc6 is seen to complete the ORC ring as expected by binding in between Orc1 and Orc2 (Fig. 1b). Surprisingly, masked classification of the Cdc6 region revealed that the ORC·Cdc6 ring adopts two distinct conformations that differ in pitch and the positioning of Cdc6 (Fig. 1c, d). The two states arise from a rocking motion of mainly Orc1 and Cdc6, which leads to ring flattening as the complex transitions from state 1 to state 2 (referred to as ODC1 and ODC2 hereafter, Fig. 1d). Both states also differ from the ring conformation seen in a loading intermediate containing DNA, ORC, Cdc6, Cdt1, and Mcm2-7 (OCCM)[8], indicating this change

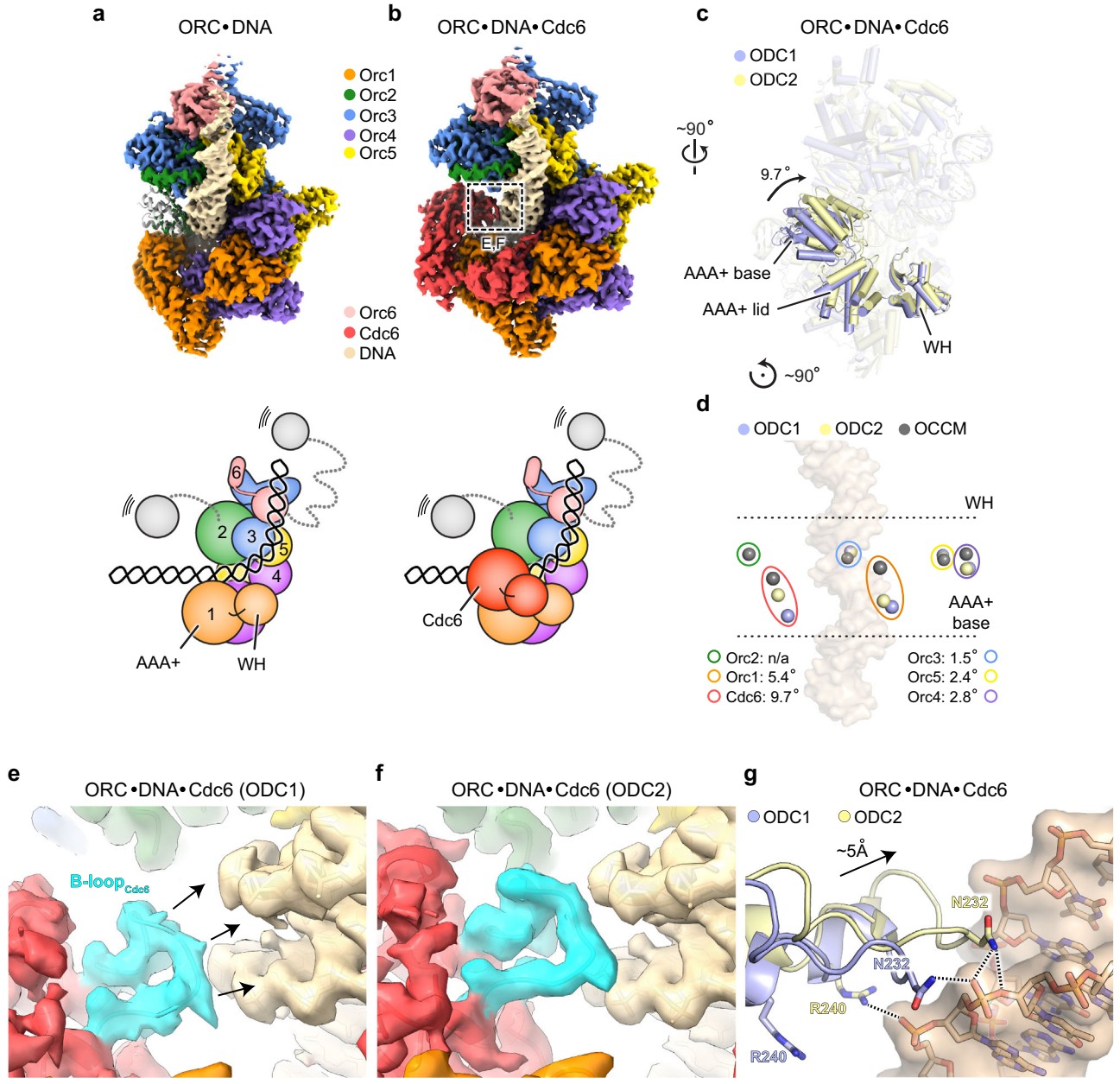

**Fig. 1 Conformational changes in the *Sc*ORC·Cdc6 ring alter Cdc6's DNA binding mode.** Cryo-EM structures of ARS1 DNA-bound *Sc*ORC (in **a**) and in complex with *Sc*Cdc6 (ODC1, in **b**). Unsharpened cryo-EM maps are shown surface-colored by subunit with schematics below. In **a**, the Orc2 WH domain of an *Sc*ORC·DNA structure obtained from crosslinked sample (PDB 5zr1[6]) is shown as gray cartoon. Weak density in the current *Sc*ORC·DNA cryo-EM map indicates flexibility of this region in the uncrosslinked complex. Zoomed views of boxed region (in **b**) are provided in panels **e** and **f**. **c** Cdc6 is found in two different positions in the ORC·Cdc6 ring, which are related by a ~10° rotation of the subunit. Both *Sc*ORC·DNA·Cdc6 structures (ODC1 in blue and ODC2 in yellow) were aligned by superposing Orc2. Cdc6 is shown as solid cartoon while other subunits are transparent (WH – winged-helix domain). **d** Repositioning of Cdc6 is caused by movement of both Orc1 and Cdc6 in the ORC·Cdc6 ring. Centers of mass of the AAA+ base domains of Orc1-5 and Cdc6 in ODC1, ODC2, and OCCM (PDB 5v8f[8]) are shown as blue, yellow, and gray spheres, respectively. Corresponding subunits are outlined using colors as in (**a**). Angular displacements of corresponding subunits between ODC1 and ODC2 are listed. **e**–**g** Cdc6 repositioning facilitates novel contacts between the Cdc6-B-loop and DNA. Unsharpened cryo-EM map densities of DNA and the Cdc6 B-loops (highlighted in cyan) of ODC1 (in **e**) and ODC2 (in **f**) are shown. In **g**, the Cdc6 B-loop regions of ODC1 and ODC2 are superposed after structural alignment of the DNA duplexes. Electrostatic interactions are indicated by dashed lines. The color schemes used in this figure are maintained throughout the manuscript unless noted otherwise.

in ring conformation may be linked to the functional ORC·Cdc6 cycle during Mcm2-7 loading (Fig. 1d).

Interestingly, the two *Sc*ORC·DNA·Cdc6 structures we determined differ in the DNA binding mode of the Cdc6 subunit. In ODC2, Cdc6, and in particular the B-loop element in its AAA + domain, a loop region that is located after the conserved catalytic Walker B motif, shifts by ~5 Å and increases its contact

area with DNA (Fig. 1e–g). The rearrangement of Cdc6 results in an additional hydrogen bond between Asn 232 and the sugar-phosphate backbone, and Arg 240 moves closer to form a salt bridge with a phosphate in the ACS (Fig. 1g). These observations are in accord with biochemical and structural findings reported for *Drosophila* ORC and Cdc6, where the B-loop elements of Orc1 and Cdc6 engage the DNA duplex and are important for

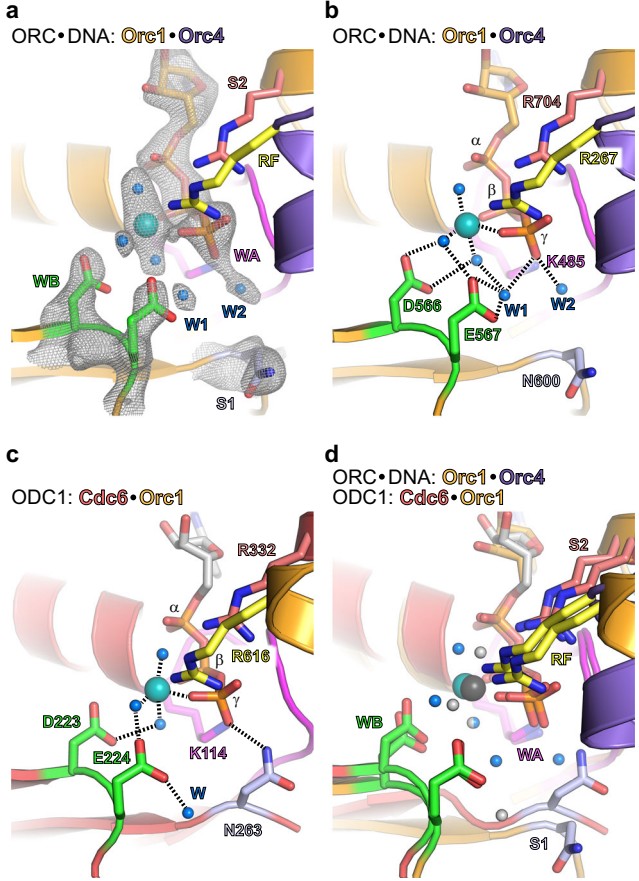

**Fig. 2 The ATPase sites reside in a pre-hydrolysis state. a** and **b** Configuration of the Orc1·Orc4 ATPase site in DNA-bound ScORC. The density-modified cryo-EM map is shown as gray mesh for ATP, Mg$^{2+}$, waters, and Walker B and sensor 1 side chains in **a**. **c** Configuration of the Cdc6·Orc1 ATPase site in ODC1. Dashed lines in **b** and **c** summarize bonding interactions of water molecules and sensor 1. Superposition of the Orc1·Orc4 and Cdc6·Orc1 ATPase sites of ScORC·DNA and ScORC·DNA·Cdc6, respectively, highlights the different conformations of the sensor 1 residues. Magnesium and waters are shown in teal and blue, respectively; an exception is the Orc1·Cdc6 site in (**d**), where the corresponding elements are colored gray. See also Supplementary Fig. 6. WA – Walker A, WB – Walker B, S1 – sensor 1, S2 – sensor 2, RF – arginine finger, W - water.

stabilizing the initiator on DNA[33]. Collectively, these results establish that the B-loop element is a bona fide DNA binding element in the AAA+ domain of eukaryotic Cdc6 co-loaders.

**The Orc1·Orc4 and Cdc6·Orc1 ATPase sites reside in a pre-hydrolysis state**. The initiator ORC contains one functional ATPase center that is formed at the interface between Orc1 and Orc4 protomers[26,47]. A second ATPase site is formed by Orc1 and Cdc6 upon recruitment of the co-loader[29,48]. The ATPase activity in ScORC·DNA and ScORC·DNA·Cdc6 is kept low through inhibition by DNA binding and the absence of stimulatory factors such as the Mcm3-WH domain[24,27]. In prior structures of yeast and metazoan initiator-containing assemblies, these interfaces formed closed ATPase sites with a trans-acting arginine finger, a residue essential for catalysis in AAA+ ATPases, engaging the ATP γ-phosphate, indicating these sites were primed for ATP hydrolysis[6,8,9,32–35]. However, the structural features of the Orc1·Orc4 ATPase center in previous models fail to explain the suppressed ATPase activity of ORC upon engaging DNA. Our 2.55 Å resolution cryo-EM map of

ScORC·DNA afforded the location of active-site water molecules (which were not resolved in any of the previous eukaryotic initiator structures due to lower resolutions) and a better definition of the side chain positions of active site residues (Fig. 2a). As expected, the conserved Walker A lysine of Orc1 (K485) bonds with the β- and γ-phosphate, while the Orc4 arginine finger (R267) and Orc1 sensor 2 arginine (R704) engage the γ-phosphate (and also the α-phosphate in case of the sensor 2). The Walker B aspartate (Orc1-D566) binds magnesium through two water molecules (Fig. 2b). The catalytic glutamate (Orc1-E567), a residue that activates a lytic water for nucleophilic attack of the γ-phosphate during catalysis, points towards the ATP γ-phosphate, its position stabilized by tether residues in Orc4, Y232, and R263[32], and a bonding interaction with a water molecule in the magnesium coordination sphere. Surprisingly, the side chain of sensor 1 (Orc1-N600), a conserved polar residue in AAA+ ATPases that supports ATP hydrolysis by helping to activate the lytic water and/or by bonding with the ATP γ-phosphate[15,49,50], is flipped away from ATP (Fig. 2a, b). Apart from the magnesium-coordinating waters, two additional ordered water molecules are observed in the active site in a cavity near the γ-phosphate (W1 and W2, Fig. 2a, b). W1 hydrogen bonds with the catalytic glutamate, while both W1 and W2 also directly bond with the γ-phosphate. However, neither of the ordered waters is correctly positioned for an in-line attack on the γ-phosphate. Together, these findings argue that the Orc1·Orc4 ATPase in DNA-bound ScORC prefers to reside in a pre-hydrolysis conformation. The presence of two active-site waters could suggest that ORC may employ a "multiple"-water mechanism for catalysis, which has been suggested for other nucleotide-hydrolyzing systems and appears energetically favorable over a single-water mechanism[51,52].

How is the Orc1·Orc4 ATPase site then activated for catalysis? Mutagenesis of the Orc1-sensor 1 in budding yeast has shown that a polar residue in this position is important for cell viability, consistent with a role of Orc1-N600 in ATP hydrolysis[53]. Thus, we predict that the following rearrangements are needed for ATP hydrolysis: (1) a rotamer switch of the sensor 1 residue, (2) repositioning of at least one of the ordered water molecules in the active site for an in-line attack, and (3) a slight shift of the Orc4-R263 tether may also be needed. The Orc4-R263 tether may fulfill a role analogous to the arginine coupler in E. coli DnaC, which has been implicated in active site regulation of the bacterial helicase loader[54]. In the DNA-free DnaBC complex, the arginine coupler DnaC-R216 occupies the site of the lytic water prior to ATP hydrolysis; likewise, Orc4-R263 may impede correct positioning of the water molecule for catalysis (Supplementary Fig. 5). One possibility is that Cdc6 binding drives ScORC into a hydrolysis-competent conformation; however, inspection of the Orc1·Orc4 ATPase site configuration in our 2.5 and 2.7 Å cryo-EM structures of ODC1 and ODC2 reveals that the active site waters remain in a non-lytic position (Supplementary Fig. 6a–c). The Orc1-sensor 1 is also flipped away from ATP in ODC1, although it appears to move closer to ATP in ODC2 (Supplementary Fig. 6a–c). Regardless, the Orc1·Orc4 ATPase centers in all three states (OD, ODC1, and ODC2) are similar and preferentially persist in a pre-hydrolysis state. Thus, ORC's ATPase activation is likely coupled to a later event during Mcm2-7 loading to allow the release of ORC from DNA and recycling of the initiator[26,55]. Indeed, ORC's ATPase activity is not required for Mcm2-7 loading in vitro but is essential for ORC function in vivo[22,26,56]. Higher resolution structures of Mcm2-7-containing loading intermediates than currently available[8,9,57] will be required to define the alterations that trigger ATP hydrolysis by ORC.

The second ATPase center in the ternary ScORC·DNA·Cdc6 complex at the interface between Orc1 and Cdc6 likewise appears to reside in a pre-hydrolysis state. Overall, the Cdc6·Orc1 sites in

ODC1 and ODC2 are configured very similarly to each other and to that of Orc1 and Orc4, but the Cdc6-sensor 1 (N263), which is important for budding yeast viability and ATP hydrolysis by Cdc6[48,58,59], points into the active site and bonds with the γ-phosphate (Fig. 2c, d and Supplementary Fig. 6d–f). A water molecule positioned for an in-line nucleophilic attack, however, is not seen. We postulate that the lytic water will be stabilized for nucleophilic attack upon docking of the winged-helix domain of Mcm3 onto ORC·Cdc6, which binds to the Cdc6 ATPase in the OCCM complex and has been shown to stimulate ATP hydrolysis of the ternary complex by several fold[8,24]. Considering the quality control function of the Cdc6 ATPase, it seems advantageous to keep its basal activity in ORC·Cdc6 low; hyperactivity would likely interfere with productive Mcm2-7 loading by prematurely releasing loading factors[21,22,24]. Intriguingly, Cdc6 binding to ScORC causes conformational rearrangements in the B-loop of Orc1 that help reorient the Orc1 arginine finger helix (with arginine finger R616) for completion of the Cdc6 ATPase site (Supplementary Fig. 7). In Drosophila ORC, the Orc1-B-loop region has been identified as an element that contributes to the coordination of DNA-binding and ATP hydrolysis in the initiator and may help with inter-ATPase site communication[33]. Combined, these observations suggest that coupling conformational changes in the Orc1 B-loop to ATPase site configuration may be a universal feature of eukaryotic initiator-co-loader assemblies.

**The N-terminal cyclin box domain of Orc6 docks onto the ORC·Cdc6 ring.** During data processing of the ternary complex, we noticed that the cryo-EM maps contained weak, fragmented density at the cleft formed by the winged-helix and AAA+ domain tiers near the Orc1 subunit (Supplementary Fig. 2c). Masked classification revealed that, in about half of the ScORC·DNA·Cdc6 particles, an additional domain composed of several α-helices was docked onto the ORC·Cdc6 ring, which has not been previously observed in the ternary complex. With a resolution range of 2.9–3.5 Å, this region of the cryo-EM map was sufficiently resolved for de novo model building (Supplementary Figs. 3f, 4b). To our surprise, we identified this region as the N-terminal cyclin box fold (CB$_N$) of Orc6 (Fig. 3a, b). Orc6-CB$_N$ binds in a groove formed by the lid sub-domain of the Orc1-AAA+ region and the winged-helix fold of Cdc6 (Fig. 3a, c). The binding of Orc6-CB$_N$ appears to stabilize the Cdc6-WH fold since the cryo-EM map shows higher resolution features in this region compared to the one without a docked Orc6-CB$_N$. The tripartite binding site buries ~990 Å$^2$ of solvent-accessible surface area and is stabilized by hydrogen bonds and salt bridges formed between Orc6-CB$_N$ and the Orc1-AAA+ lid, and by van der Waals interactions between Orc6-CB$_N$ and Cdc6-WH (Fig. 3d). This bonding pattern explains why Orc6-CB$_N$ is not seen docked onto the ORC ring in ScORC·DNA without Cdc6.

In contrast to Orc6-CB$_N$, the C-terminal Orc6 cyclin box fold (CB$_C$) is bound at the opposite side of the ORC·Cdc6 ring and is sandwiched between the Orc3 insertion domain, the winged-helix domains of Orc3 and Orc5, and the bent DNA duplex as seen in previous structures of the yeast initiator and other loading intermediates[6,9,34,57] (Fig. 3a). This configuration separates the C-terminal residue of Orc6-CB$_N$ and the N-terminal residue of Orc6-CB$_C$ by 123 Å, which can easily be bridged by the 156 amino acid-long, flexible linker that connects both domains but is not resolved in our cryo-EM maps (Fig. 3a, b). Further subclassification of Orc6-CB$_N$ particles revealed that they can also adopt both ODC1 and ODC2 ring conformations, indicating that Orc6-CB$_N$ docking does not alter the equilibrium between both states, nor does it change the configuration of the ATPase centers (Supplementary Fig. 8a).

To confirm the interaction between the N-terminal domain of Orc6, Orc1, and Cdc6 with an independent technique, we turned to crosslinking mass spectrometry. Using the cleavable crosslinker DSSO, we identified several crosslinks between lysines in the Orc6-CB$_N$ or the proximal Orc6 linker region (immediately following the CB$_N$ up to residue 150) and the Cdc6-WH domain or the Orc1-AAA+ lid region (Fig. 3e and Supplementary Fig. 8b, c). The Orc1-WH domain also crosslinked to the proximal Orc6 linker and is likely located along the path the chain takes towards the Orc6-CB$_C$. By contrast, the Orc6-CB$_C$ predominantly cross-linked to Orc2 on the opposite side of the ORC·Cdc6 ring (Supplementary Fig. 8b, c). These findings agree with our cryo-EM data and establish that a docked Orc6-CB$_N$ onto the ORC·Cdc6 ring (referred to as ORC·DNA·Cdc6·Orc6-CB$_N$ here-after) is a bona fide state of the initiator·co-loader complex adopted by a subset of ternary ScORC·DNA·Cdc6 assemblies.

**ScORC·DNA·Cdc6·Orc6-CB$_N$ represents an autoinhibited state that hinders Mcm2-7 loading.** The N-terminal cyclin box fold of Orc6 has previously only been observed in one of the Mcm2-7 loading intermediates, the Mcm2-7·ORC (MO) complex, in which the first Mcm2-7 hexamer has been fully loaded onto DNA, and Cdt1 and Cdc6 have been released[57]. In the 4.4 Å structure of this intermediate, the Orc6-CB$_N$ can be seen latched onto the Mcm2-7 ring, binding to the N-terminal domains of Mcm2 and Mcm6[57], and helps keep ORC and the first Mcm2-7 hexamer in close proximity until the second hexamer has been loaded for dimerization[60] (Fig. 4a). Structural alignment of Orc6-CB$_N$ as seen in the MO structure and in our structure revealed that the overall fold of the N-terminal Orc6 domain is not significantly altered between both assemblies (RMSD 3.25 Å), and is also similar to that of the human counterpart[57,61] (Fig. 4b and Supplementary Fig. 4c). Intriguingly, mapping the binding sites for Mcm2/Mcm6 (in MO) and Orc1/Cdc6 (in ScORC·DNA·Cdc6·Orc6-CB$_N$) onto Orc6-CB$_N$ shows that there is a substantial overlap between the interacting regions, indicating that docking of Orc6-CB$_N$ onto Mcm2-7 and the ORC·Cdc6 ring is mutually exclusive (Fig. 4c and Supplementary Fig. 9a). Indeed, superposing Orc6-CB$_N$ in ScORC·DNA·Cdc6 with that in the MO complex results in severe clashes of the Orc1-AAA+ lid with Mcm6 and of the Cdc6-WH domain with Mcm2 (Supplementary Fig. 9b). Single-molecule studies have shown that a second Cdc6 is recruited to ORC after loading of the first Mcm2-7 hexamer[55], forming an MO·Cdc6 complex with an Orc1/Cdc6 platform that could potentially sequester Orc6-CB$_N$. Since Orc6-CB$_N$ contacts with the first loaded Mcm2-7 hexamer in the MO intermediate are key for efficient double hexamer formation[57,60], competition between the distinct Orc6-CB$_N$ binding sites could provide a means to regulate whether loading proceeds towards double hexamer formation or whether the first hexamer disengages from ORC.

How Orc6-CB$_N$ attaches to the ORC·Cdc6 ring seems reminiscent of how Mcm2-7 latches onto ORC·Cdc6 during OCCM formation[8]. In the OCCM intermediate, the C-terminal winged-helix folds of Mcm3, Mcm7, Mcm4, and Mcm6 bind to a groove formed by the AAA+ and winged-helix domain tiers of Orc1-5 and Cdc6, engaging the AAA+ lid subdomain of one subunit and the winged-helix domain of the adjacent protomer. Strikingly, structural alignment of the 3.9 Å resolution OCCM structure and our 2.7 Å ScORC·DNA·Cdc6·Orc6-CB$_N$ structure revealed that Orc6-CB$_N$ occupies the same position as the winged-helix domain of Mcm7 in OCCM (Fig. 4d, e). This arrangement of Orc6-CB$_N$ in ScORC·DNA·Cdc6 is expected to preclude productive Mcm2-7 docking and consequently loading because Mcm7 cannot engage the initiator and co-loader. Mcm2-7 attachment to ORC·Cdc6 would be stalled after binding of the

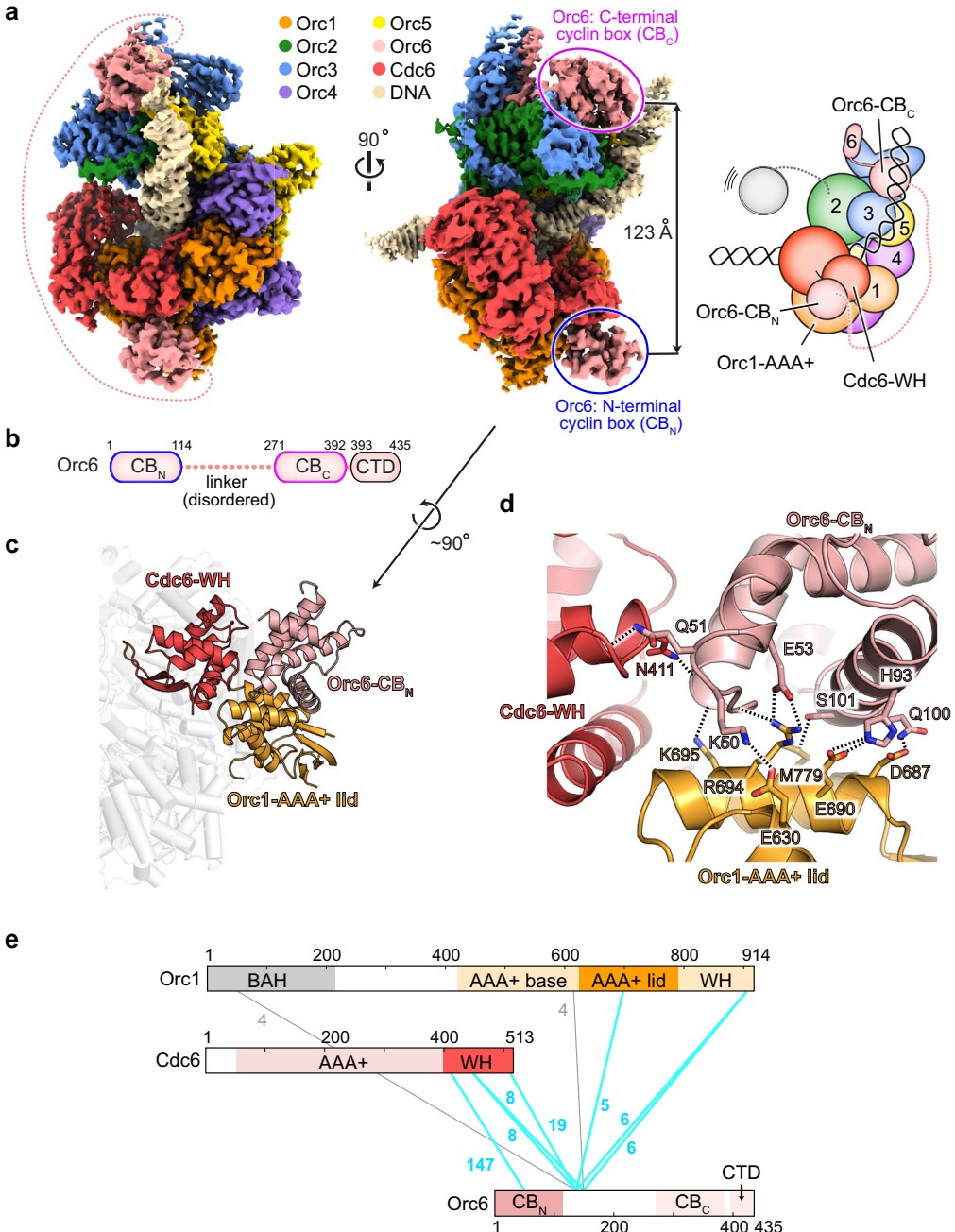

**Fig. 3 The N-terminal domain of Orc6 docks onto the ORC·Cdc6 ring through interactions with Orc1 and Cdc6. a** Unsharpened cryo-EM map and schematic of *Sc*ORC·DNA·Cdc6 with a docked N-terminal cyclin box domain (CB_N) of Orc6. The flexible linker between N- and C-terminal CB folds is indicated by a dashed line. **b** Scheme of *Sc*Orc6's domain architecture. CB_N and CB_C share structural similarity with transcription factor TFIIB, which is predicated on cyclin box folds. **c** and **d** Orc6-CB_N directly docks onto the Cdc6 WH domain and the lid subdomain of the Orc1 AAA+ fold. Interacting domains are rendered as colored cartoon in (**c**). A zoomed view of the binding site is shown in (**d**) with interacting side chain residues rendered as sticks and hydrogen bonds and salt bridges displayed by dashed lines. **e** Crosslinks between Orc1, Cdc6, and Orc6 identified by mass spectrometry. Cyan lines highlight crosslinks involving Orc6-CB_N or the proximal linker region immediately following this domain. The number of crosslink-spectra matches (CSM) is listed for each crosslink. See Supplementary Fig. 8c for a summary of observed crosslinks between all subunits in *Sc*ORC·DNA·Cdc6. CB – cyclin box domain, CTD – C-terminal domain, WH – winged-helix domain, BAH – bromo-adjacent homology domain.

Mcm3-WH domain, which mediates the first encounter between the initiator·co-loader and Mcm2-7[24]. This partially attached OCCM intermediate is likely less stable due to fewer interactions between ORC·DNA·Cdc6 and Mcm2-7 and is consistent with biochemical findings that deletions of the *Sc*Mcm7-WH domain reduce Mcm5/3/7 trimer and Mcm2-7 hexamer recruitment to DNA[62]. Interestingly, a "semi-attached" OCCM complex has been reported in which both Mcm3-WH and Mcm7-WH folds were latched onto the ORC·Cdc6 ring, but the Mcm4 and Mcm6 winged-helix domains and the Mcm2-7 ring itself were not docked[9]. Surprisingly, comparison of the cryo-EM maps of this "semi-attached" OCCM and our assembly revealed that the map region previously assigned as Mcm7-WH corresponds to Orc6-CB_N (Supplementary Fig. 10). Since Orc6-CB_N and Mcm7-WH bind the same site on the ORC·Cdc6 ring, we posit that the *Sc*ORC·DNA·Cdc6·Orc6-CB_N intermediate we discovered is not

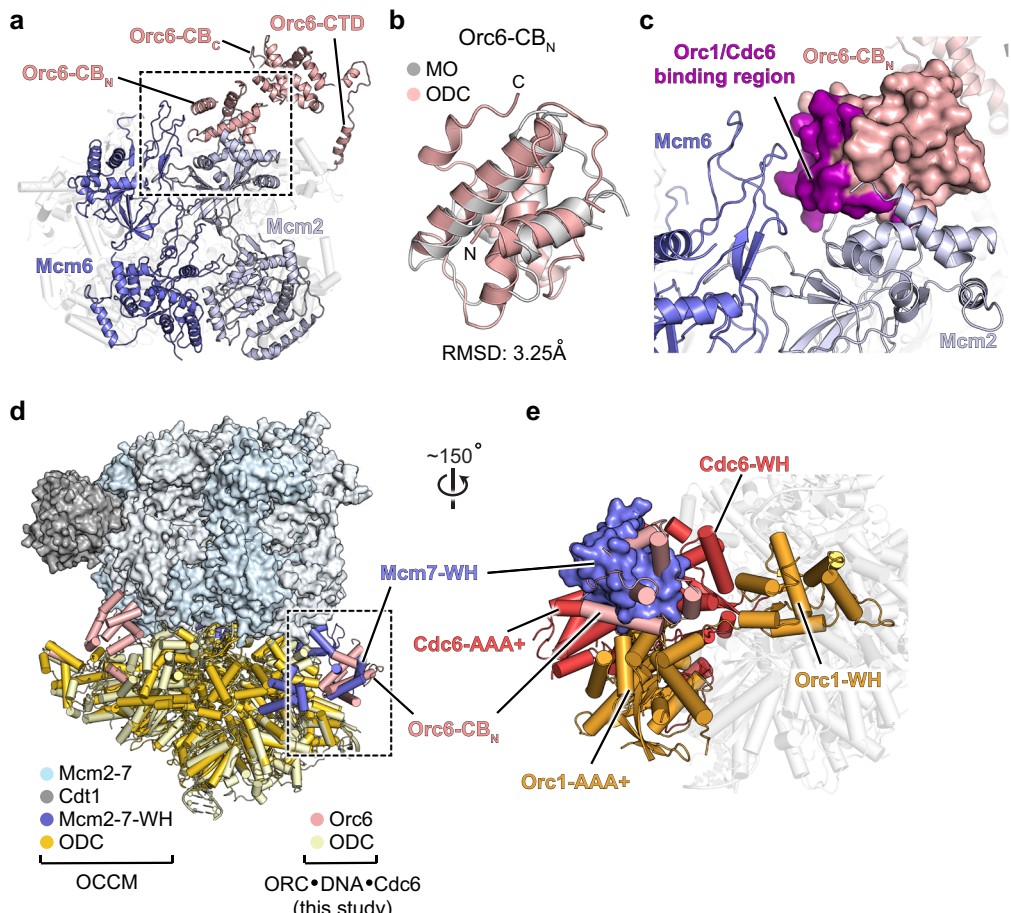

**Fig. 4 Docking of the Orc6 N-terminal domain onto the ORC·Cdc6 ring is incompatible with MO and OCCM formation. a–c** Overlapping binding sites of Orc6-$CB_N$ with Mcm2-7 in the Mcm2-7·ORC complex (MO, PDB 6rqc[57]) and with Orc1 and Cdc6 in ScORC·DNA·Cdc6. **a** Orc6-$CB_N$ binds to Mcm2 and Mcm6 in the MO complex. Mcm2, Mcm6, and Orc6 are depicted as colored cartoon, while other Mcm2-7 subunits are shown in light gray as transparent cartoon. Other ORC subunits are omitted for clarity. **b** Structural alignment of Orc6-$CB_N$ in the MO complex with this domain in ScORC·DNA·Cdc6. The root mean square deviation (RMSD) of atomic coordinates is listed. **c** Zoomed view of boxed region in **a** with Orc6-$CB_N$ from ScORC·DNA·Cdc6 (shown as surface) superposed onto the corresponding domain in the MO complex (shown as cartoon). The Orc6-$CB_N$ surface involved in interactions with Cdc6 and Orc1 in ScORC·DNA·Cdc6 is colored purple. **d** and **e** Orc6-$CB_N$ in ScORC·DNA·Cdc6 occupies a similar position as the winged-helix domain (WH) of Mcm7 in the OCCM complex. **d** Superposition of ScORC·DNA·Cdc6 (this study) and OCCM (PDB 5v8f[8]) by structural alignment of Orc1 in both complexes. Mcm2-7 and Cdt1 are represented as surface except for the Mcm2-7 WH domains (in blue cartoon), while ORC and Cdc6 are depicted as cartoon. **e** Zoomed view of boxed region from **d** highlights that the Mcm7-WH domain (as positioned in the OCCM complex) would clash with Orc6-$CB_N$ that is docked onto the ORC·Cdc6 ring. Mcm7-WH is rendered as blue surface, while other OCCM regions are omitted for clarity. ScORC·DNA·Cdc6 (this study) is shown as cartoon with Orc1, Orc6, and Cdc6 colored.

an on-pathway intermediate towards Mcm2-7 loading but instead corresponds to an autoinhibited state of the initiator·co-loader assembly that stalls helicase deposition.

**Orc6-$CB_N$ inhibits Mcm7 recruitment in response to phosphorylation.** What may be the function of an autoinhibited initiator·co-loader assembly? Given that Orc6-$CB_N$, when docked onto the ORC·Cdc6 ring, is predicted to block Mcm2-7 loading, it is plausible that the autoinhibited state is involved in regulating the timing of origin licensing. In *S. cerevisiae* cells, Mcm2-7 loading is inhibited outside of G1, a strategy that is key to preventing re-replication of DNA[36,38]. One of the mechanisms employed by *S. cerevisiae* to prevent helicase loading in this context is CDK-mediated phosphorylation of Orc2 and Orc6[24,39–41]. Interestingly, Orc6 in our ORC preparations contained a mixture of unphosphorylated and phosphorylated Orc6 and included modifications of established CDK sites[39,40], which reside in the proximal linker region (immediately following the

Orc6-$CB_N$) that we identified to crosslink to Orc1 and Cdc6 (Fig. 3e and Supplementary Figs. 1b, 8c, and 11). We thus hypothesized that posttranslational modification of phosphosites in Orc6 may promote docking of the N-terminal Orc6 domain onto the ORC·Cdc6 ring to prevent productive OCCM assembly. To test this premise, we devised an assay to monitor recruitment of Mcm7 to the initiator·co-loader complex. Mcm7 and the Orc6-$CB_N$ domain interact with the same site in ScORC·DNA·Cdc6, and both binding events are thus expected to be mutually exclusive (Fig. 5a). In the presence of ATPγS, we observed weak but reproducible co-association of Mcm7 with ScORC·DNA·Cdc6 above background levels (no ATPγS reactions) in pulldowns (Fig. 5b). The relatively weak binding is not surprising given that the interface between the Mcm7-WH domain and Orc1/Cdc6 is small and that other subunits (e.g., Mcm3) stabilize ORC·Cdc6-MCM interactions in the context of the Mcm2-7 hexamer[62]. Importantly, the amount of Mcm7 that co-purified with ScORC·DNA·Cdc6 increased after dephosphorylation of ORC with λ phosphatase but was reduced when ORC was

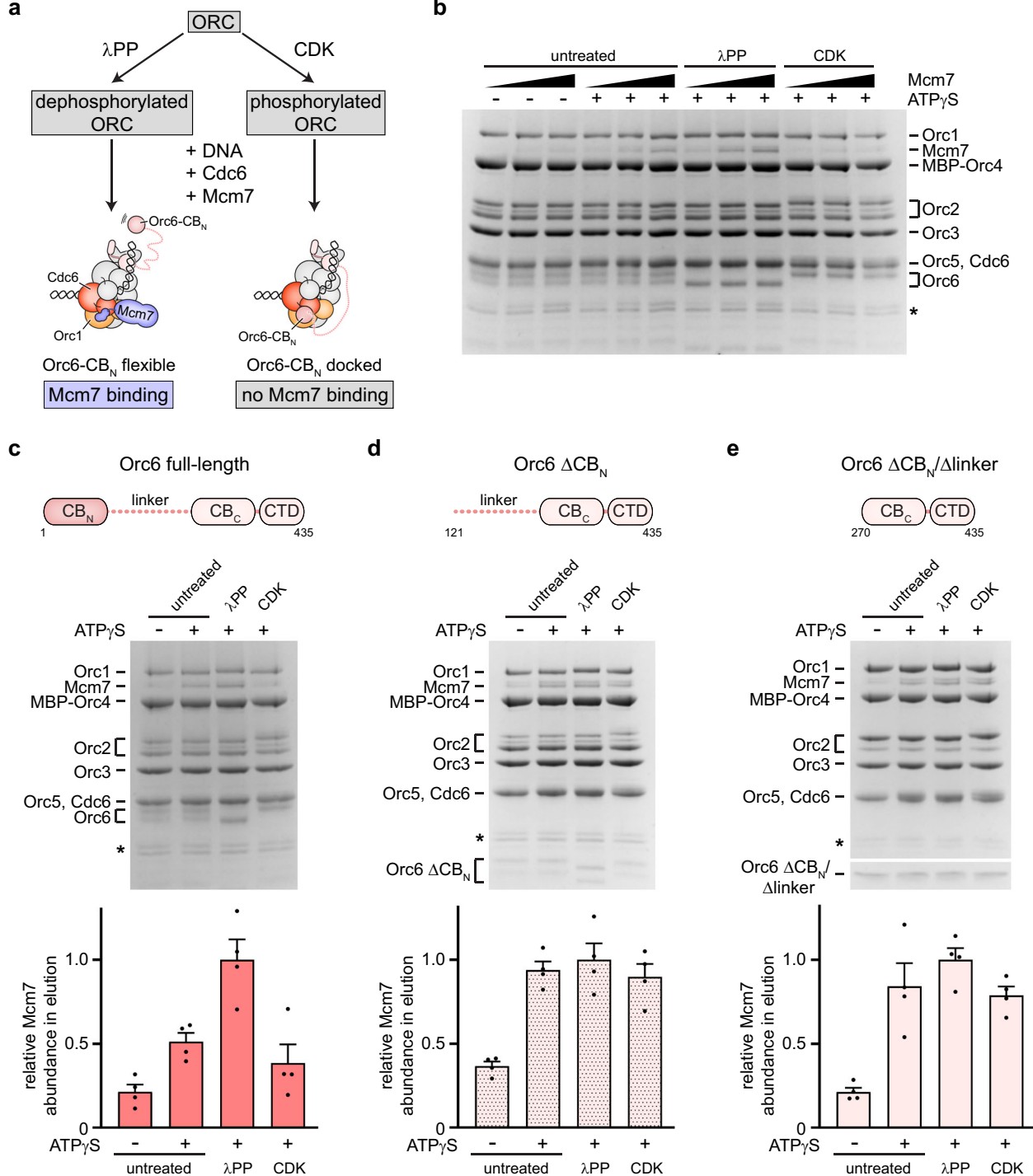

**Fig. 5 Orc6-CB_N is required for CDK-mediated inhibition of Mcm7 recruitment to ScORC·DNA·Cdc6. a** Experimental workflow of Mcm7 recruitment assay. Docking of Orc6-CB_N onto the ORC·Cdc6 ring is predicted to prevent binding of Mcm7. **b** Binding of ScMcm7 to ScORC·DNA·Cdc6 is increased after dephosphorylation and decreased upon CDK phosphorylation of ORC. Association of increasing amounts of Mcm7 with the initiator·co-loader was assessed in amylose pulldowns using ORC with MBP-tagged Orc4 as bait. **c–e** Deletion of Orc6-CB_N abrogates the phosphorylation-dependent regulation of ScMcm7 binding to ScORC·DNA·Cdc6. Coomassie-stained SDS-PAGE gels of elutions from amylose pulldowns are shown in (**b–e**). The mean and standard error of the mean of ScMcm7 band intensities ($n = 4$ independent recruitment assays) are plotted in (**c–e**). Note that Mcm7 migrates as a doublet and that Orc2 and the unstructured linker region of Orc6ΔCB_N are prone to proteolytic cleavage during expression and purification. Asterisks mark degradation products of ORC subunits that are stably integrated into ORC. In **e**, a higher percentage gel was used to visualize Orc6ΔCB_N/Δlinker due to its low molecular weight. SDS-PAGE gels of inputs from **b–e** are included in Supplementary Fig. 12. Source data are provided as a Source Data file.

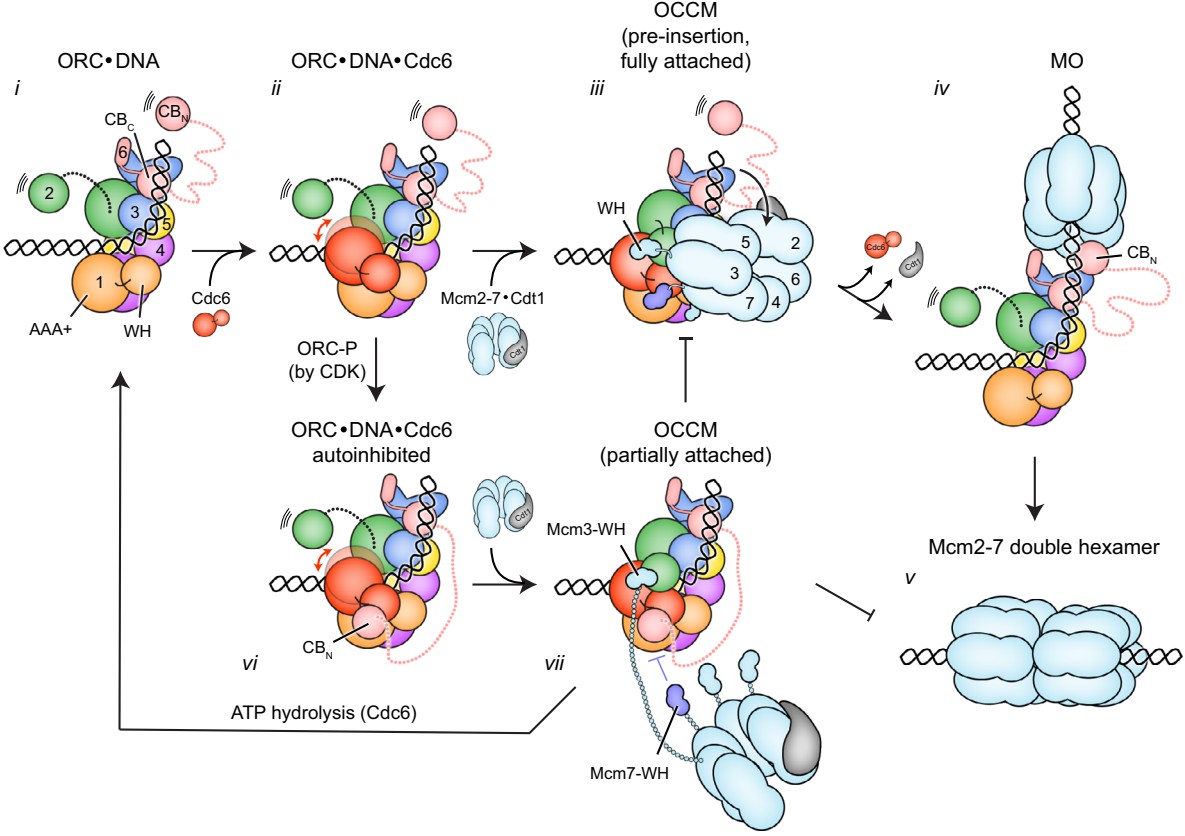

**Fig. 6 Model for the regulation of Mcm2-7 loading by the N-terminal domain of Orc6.** i ORC binds and bends DNA. The Orc6-CB$_N$ is flexibly tethered to ORC. ii Cdc6 binds to ORC·DNA. The ternary complex can adopt different ring conformations. iii Mcm2-7 and Cdt1 are recruited and fully attach to the ORC·Cdc6 ring. The bent DNA is positioned near the Mcm2/5 gate. iv The first Mcm2-7 hexamer is loaded onto DNA and held in proximity to ORC by the Orc6-CB$_N$ in the MO complex. Cdc6 and Cdt1 have been released. v The MO complex recruits a second Cdc6, Cdt1, and Mcm2-7 complex to load another hexamer by a similar mechanism, yielding double Mcm2-7 hexamers. vi Orc6-CB$_N$ docking onto the ORC·Cdc6 ring yields an autoinhibited ORC·DNA·Cdc6 complex. vii Mcm2-7 attachment is stalled after Mcm3-WH binding. Mcm7-WH cannot engage ORC·Cdc6 because its binding site is occupied by Orc6-CB$_N$. Helicase loading is blocked and ATP hydrolysis by Cdc6 disassembles the stalled intermediate.

phosphorylated by CDK (Fig. 5b). If these differences in Mcm7 recruitment are indeed coupled to phosphorylation-induced docking of Orc6-CB$_N$ onto the ORC·Cdc6 ring, then deletion of this domain should abrogate the phosphoregulation of Mcm7 recruitment. Congruent to this prediction, ORC assemblies that lacked the N-terminal Orc6 cyclin box fold, alone (Orc6 ΔCB$_N$) or in combination with the flexible linker region (Orc6 ΔCB$_N$/Δlinker), recruited similar amounts of Mcm7 irrespective of phosphatase or kinase pretreatment (Fig. 5c–e). These results are consistent with an autoinhibitory role of Orc6-CB$_N$ during Mcm2-7 recruitment in response to ORC phosphorylation.

Collectively, our biochemical and structural findings support a revised model for the regulation of Mcm2-7 loading (Fig. 6). Cdc6 binding to DNA-bound ORC completes the AAA+ ATPase ring, which can adopt two distinct conformations that are characterized by repositioning of Cdc6. In both states, the predominant pre-hydrolysis configuration of the ATPase sites would maintain the low basal ATPase activity of ORC·Cdc6, which likely prevents premature disassembly of the heptameric intermediate. When Orc6 is not phosphorylated (under conditions of low CDK activity), Orc6-CB$_N$ is flexibly tethered to ORC and the ORC·Cdc6 ring is available for full Mcm2-7 attachment in the pre-insertion OCCM[9] and subsequent loading of the first Mcm2-7 hexamer onto DNA. By contrast, high CDK activity leads to ORC phosphorylation and autoinhibition of ORC·DNA·Cdc6 through docking of the Orc6-CB$_N$ onto the initiator·co-loader ring, either *in cis* (involving the Orc6-CB$_N$ of the same ORC) or

in trans (involving the Orc6-CB$_N$ of a second ORC). In both cases, Mcm2-7 recruitment to autoinhibited ORC·DNA·Cdc6 would stall at a partially attached OCCM stage[9], stimulating Cdc6 ATPase activity (through Mcm3-WH binding), OCCM disassembly, and Mcm2-7/Cdt1 release as part of the quality control pathway[21,22,24] (Fig. 6). Such a mechanism may apply to the loading of the first and second Mcm2-7 hexamer, as recruitment of both occurs through a similar ORC/Cdc6-mediated mechanism[24,57,60,63]. Additionally, an autoinhibited ORC·DNA·Cdc6 state could also impede loading by sequestering Orc6-CB$_N$ from other partner proteins, for example, Cdt1[39,64] or Mcm2/6 in MO[57] (after association of the second Cdc6[55]), which could potentially uncouple the loading of the first and second hexamer. Within ORC, both Orc2 and Orc6 are substrates for CDK phosphorylation[39,40]. Future studies are needed to address whether phosphorylation of Orc6 alone or of both Orc2 and Orc6 contributes to ORC·DNA·Cdc6 autoinhibition; however, because of the proximity of the Orc6 phosphorylation sites to the Orc6-CB$_N$, we favor a model wherein autoinhibition is regulated by Orc6 phosphorylation alone and Orc2 phosphorylation prevents origin licensing through a distinct but complementary mechanism.

At this point, it is unknown whether autoinhibition of ORC·DNA·Cdc6 extends to other eukaryotic systems, but several observations argue that this state may be limited to *S. cerevisiae* and closely related fungal species. First, structural studies of *Drosophila* ORC·DNA·Cdc6 have so far not resolved the

Orc6-CB$_N$ or -CB$_C$[33]. Second, human Orc6 only weakly associates with the Orc1-5 subunits[65–67]. Third, the linker region joining Orc6-CB$_N$ and Orc6-CB$_C$ is substantially shorter in metazoan and in *S. pombe* Orc6 orthologs and not long enough to reach across the ORC·Cdc6 ring, although binding of Orc6-CB$_N$ in trans to a second ORC may be possible[67,68] (Supplementary Fig. 11). Not surprisingly, these systems also do not rely on Orc6 phosphorylation but employ other strategies to prevent Mcm2-7 loading in S phase and G2[37,38]. Nonetheless, it will be interesting to see whether functionally related inhibitory states that help to temporally coordinate the various steps involved in Mcm2-7 double hexamer formation can be identified in metazoans.

## Methods

**Expression and purification of *S. cerevisiae* ORC and Cdc6.** Wild-type *Saccharomyces cerevisiae* (*Sc*) Orc1-6 subunits and Cdc6 were cloned into pFastBac-derived baculovirus expression vectors (series 4 vectors, QB3 MacroLab, UC Berkeley) by ligation independent cloning (LIC). *Sc*Orc1 and *Sc*Orc4 were N-terminally tagged with hexa-histidine (6xHis) and maltose binding protein (MBP), respectively, while Cdc6 was fused to an N-terminal 6xHis-MBP tag. All affinity tags were followed by a Tobacco Etch Virus (TEV) protease cleavage site. For purification of MBP-ORC, full-length Orc6 and truncations (Orc6Δ1-120 and Orc6Δ1-269) were generated as C-terminal 6xHis fusions without a TEV site.

Full-length *Sc*ORC was expressed in High5 insect cells using the Bac-to-Bac Baculovirus Expression System (Thermo Fisher Scientific). Individual baculoviruses for each ORC subunit and Cdc6 were generated in Sf9 cells by transfection of bacmids isolated from DH10Bac *E. coli* with Cellfectin II (Thermo Fisher Scientific). After amplification in Sf9 cells, viruses were used to co-infect 4 L High5 cell cultures for *Sc*ORC expression. Two days later, High5 cells were harvested by centrifugation and the cell pellet was resuspended in ~140 mL lysis buffer (50 mM Tris-HCl (pH 7.8), 300 mM KCl, 10% glycerol, 50 mM imidazole, 1 mM ß-ME, 200 µM PMSF, 1 µg/mL leupeptin). The cell suspension was sonicated and the lysate clarified by ultracentrifugation at 142,032 × g for 45 min (Beckman Coulter Optima L-80 XP ultracentrifuge). The soluble fraction was subjected to ammonium sulfate precipitation (final 20% (v/v)) and subsequently cleared once more by ultracentrifugation. The supernatant was loaded onto a 5 mL HisTrap HP Nickel-affinity chromatography column (Cytiva) that was washed with 60 mL lysis buffer prior to *Sc*ORC elution with a 50–250 mM imidazole gradient. Peak fractions were further purified on 7.5–10 mL amylose columns (New England Biolabs) in 50 mM Tris-HCl (pH 7.8), 300 mM KCl, 10% glycerol, 1 mM ß-ME and eluted with 20 mM maltose. *Sc*ORC was incubated with 6xHis-tagged TEV protease overnight and then purified over another 5 mL HisTrap HP column (Cytiva, equilibrated in 50 mM Tris-HCl (pH 7.8), 300 mM KCl, 50 mM imidazole, 1 mM ß-ME) to remove the protease. The flow-through was concentrated and loaded onto a HiPrep 16/60 Sephacryl S-400 HR column (Cytiva) equilibrated in 25 mM HEPES-KOH (pH 7.6), 300 mM KCl, 10% glycerol, 1 mM DTT. *Sc*ORC peak fractions were pooled and concentrated in a 30 K Amicon Ultra-15 concentrator (Millipore), aliquoted, and flash-frozen in liquid nitrogen for storage at −80 °C. For purification of MBP-ORC, TEV cleavage and the subsequent Nickel affinity step were omitted. Full-length *Sc*Cdc6 was expressed and purified as described for *Sc*ORC with the exception that all buffers were supplemented with 10 mM magnesium acetate and a HiLoad 16/600 Superdex 200 pg column (Cytiva) was used for final gel filtration chromatography. All purification steps were done at 4 °C.

**Expression and purification of *S. cerevisiae* Mcm7.** *Sc*Mcm7 was cloned as N-terminal 6xHis-MBP-TEV fusion into a LIC-converted, pET-derived *E. coli* expression vector (Macrolab, University of California Berkeley, USA) and expressed in BL21 RIL *E. coli* cells (cultured in 4 L 2xYT medium) upon induction with 0.5 mM isopropyl β-D-1-thiogalactopyranoside (IPTG) overnight at 16 °C. Cells were harvested by centrifugation and resuspended in 70 mL lysis buffer (25 mM Tris-HCl (pH 7.8), 1 M NaCl, 10% glycerol, 1 mM ß-ME, 200 µM PMSF, 1 µg/mL leupeptin). After sonication, the lysate was clarified by centrifugation at 23,426 × g in a Sorvall Evolution RC Superspeed centrifuge (Thermo Fisher Scientific). The supernatant was loaded onto a 5 mL HisTrap HP column (Cytiva), washed with 300 mL 25 mM Tris-HCl (pH 7.8), 1 M NaCl, 20 mM imidazole, 10% glycerol, 1 mM ß-ME, and eluted with 250 mM imidazole. *Sc*Mcm7 was further purified on a 7 mL amylose column (New England Biolabs) in 25 mM Tris-HCl (pH 7.8), 400 mM NaCl, 10% glycerol, 1 mM ß-ME and eluted with 20 mM maltose. Following overnight incubation with 6xHis-tagged TEV protease, the protein solution was passed over another 5 mL HisTrap HP column (Cytiva) to remove TEV and 6xHis-MBP. The flow-through was concentrated and loaded onto Superose 6 Increase 10/300 GL or HiLoad 16/600 Superdex 200 pg columns (Cytiva) equilibrated in 25 mM HEPES-KOH (pH 7.5), 300 mM potassium acetate, 10% glycerol, 1 mM DTT. *Sc*Mcm7 peak fractions were pooled and concentrated in a 30 K Amicon Ultra-15 concentrator (Millipore), aliquoted, and flash-frozen in liquid nitrogen.

**Expression and purification of *S. cerevisiae* CDK.** Full-length *Sc*Cdc28 and N-terminally truncated Clb5 (ΔN1-94[69]) were cloned into a multibac baculovirus expression vector (series 11 vectors, QB3 MacroLab, UC Berkeley) as N-terminal 6xHis-TEV and MBP-TEV fusions, respectively, by LIC and BioBrick cloning. Both proteins were co-expressed in 2 L High5 cells infected with multibac virus generated in Sf9 cells according to the Bac-to-Bac Expression System (Thermo Fisher Scientific). 44-48 h after infection, High5 cells were harvested and resuspended in ~70 mL lysis buffer (50 mM Tris-HCl (pH 7.8), 300 mM KCl, 10% glycerol, 30 mM imidazole, 1 mM ß-ME, 200 µM PMSF, 1 µg/mL leupeptin). The cell suspension was sonicated and the lysate clarified by two rounds of ultracentrifugation as described for ORC. CDK was purified on a 5 mL HisTrap HP column (Cytiva) using a 300 mL wash with lysis buffer prior to *Sc*CDK elution with a 50–250 mM imidazole gradient (in lysis buffer without protease inhibitors). Subsequently, eluted *Sc*CDK was bound to 2 mL amylose resin (New England Biolabs), washed with 25 mL buffer (50 mM Tris-HCl (pH 7.8), 300 mM KCl, 10% glycerol, 1 mM ß-ME), and eluted with 20 mM maltose. *Sc*CDK was concentrated and further purified by gel filtration chromatography on a Superose 6 Increase 10/300 GL column (Cytiva) equilibrated in 50 mM Tris-HCl (pH 7.8), 300 mM KCl, 10% glycerol, 1 mM DTT. Protein peak fractions were pooled, concentrated in a 10 K Amicon Ultra-4 concentrator (Millipore), aliquoted, and flash-frozen in liquid nitrogen.

**Reconstitution and purification of *S. cerevisiae* ORC·DNA·Cdc6.** *Sc*ORC and *Sc*Cdc6 were dialyzed overnight at 4 °C into assembly buffer containing 25 mM HEPES-KOH (pH 7.6), 250 mM potassium acetate, 10 mM magnesium acetate, 1 mM DTT. An 84 bp ARS1 DNA fragment (5′-TTTGTGCACTTGCCTGCAG GCCTTTTGAAAAGCAAGCATAAAAGATCTAAACATAAAATCTGTAAAAT AACAAGATGTAAAGAT-3′ and 5′-ATCTTTACATCTTGTTATTTTACAGAT TTTATGTTTAGATCTTTTATGCTTGCTTTTCAAAAGGCCTGCAGGCAAG TGCACAAA-3′ annealed at 50 µM in 10 mM Tris-HCl (pH 8), 5 mM MgCl$_2$) was then mixed with *Sc*ORC in assembly buffer supplemented with 1 mM ATP. After 5 min incubation at room temperature, *Sc*Cdc6 was added, incubated for another 5 min, and transferred to 4 °C. The final reaction (300 µL total volume) contained 3 µM DNA, 2.5 µM *Sc*ORC, and 6.8 µM *Sc*Cdc6. and was purified on a Superose 6 Increase 10/300GL column (Cytiva) equilibrated in 25 mM HEPES-KOH (pH 7.6), 250 mM potassium acetate, 10 mM magnesium acetate, 1 mM DTT, 0.2 mM ATP. Peak fractions containing the ternary *Sc*ORC·DNA·Cdc6 complex were pooled and concentrated in a 30 K Amicon Ultra-4 concentrator to an absorbance (280 nm) of 1.8.

**Cryo-EM data collection and image processing.** Cryo-EM grids were prepared by applying 3.5 µL of concentrated *Sc*ORC·DNA·Cdc6 complex (supplemented with 0.001% TWEEN20) to 300-mesh R1.2/1.3 UltrAuFoil grids (Quantifoil Micro Tools GmbH), which had been freshly plasma-cleaned for 2 min at 5 W (18.9% H$_2$ /81.1% O$_2$ gas mixture). The sample was incubated on the grid for 10 s, blotted for 2 s and vitrified in liquid ethane using a Vitrobot Mark IV plunge freezer (Thermo Fisher Scientific). Cryo-EM data were recorded as dose-fractionated movies on a Titan Krios G2 cryo-electron microscope (Thermo Fisher Scientific) at an acceleration voltage of 300 kV using a post-GIF (Quantum LS imaging filter with a slit width of 20 eV) and a Gatan K2 summit direct electron detector. The microscope was also equipped with a spherical aberration corrector (CEOS GmbH Heidelberg, Germany). Automated data collection was performed with EPU software (Thermo Fisher Scientific) using fringe-free illumination with a beam-size of 750 nm and 3 exposures per 1.2 µm hole. The targeting defocus was set to −0.8 to −1.6 µm. Movies were recorded at a pixel size of 0.86 Å as 50 frames for a total of 7.5 s with a dose rate of 6.6 e⁻/Å² per second, yielding a total electron dose of 49.3 e⁻/Å².

For image processing, dose-fractionated movies were first motion-corrected with MotionCor2[70], and contrast transfer function parameters were determined with GCTF using non-dose-weighted, motion-corrected sums of movie frames[71]. Particles were auto-picked with GAUTOMATCH[72] from the dose-weighted movie-frame sums using low-pass filtered (to 30 Å) class averages obtained from a smaller test dataset as templates. Picked particles were extracted and normalized in RELION 2.1[73,74] with a box size of 300 × 300 pixels at a pixel size of 0.86 Å.

Extracted particles were further processed in RELION 2.1[74] and RELION 3.0[75]. 3D classification into four classes using a map of a previously recorded *Sc*ORC·DNA·Cdc6 test dataset (low-pass filtered initially to 60 Å) as a reference yielded one class with clearly resolved secondary structure elements that were refined using the autorefinement procedure in RELION. Visual inspection of the map showed that the density for Cdc6 was weaker than for ORC subunits, indicating that Cdc6 was substoichiometric or adopted multiple conformations. To sort particles further, the refined particle set was subjected to 3D classification within a soft-edged Cdc6 mask (generated from a *Drosophila* Cdc6 structure[33] docked into the cryo-EM map) without particle alignment. Particles in each class were subjected to 3D refinement, which yielded one map for *Sc*ORC·DNA (without Cdc6) and two maps for *Sc*ORC·DNA·Cdc6 that differed in the conformation of Cdc6 in the ORC·Cdc6 ring (states I (ODC1) and II (ODC2)). Both *Sc*ORC·DNA·Cdc6 maps contained additional weak and fragmented density near the Orc1 and Cdc6 subunits. Masked classification using a soft-edged mask encompassing this density allowed for sorting of *Sc*ORC·DNA·Cdc6 particles (after aligning all *Sc*ORC·DNA·Cdc6 particles to a consensus *Sc*ORC·DNA·Cdc6 map) into two groups that were further refined, one that contained a well-ordered N-

terminal cyclin box domain of Orc6 docked onto the ORC·Cdc6 ring and another where this domain was flexible. Further subclassification and refinement revealed that the ternary complex with a docked Orc6 N-terminal domain could be subdivided into ODC1 and ODC2 states based on different conformations of Cdc6. Bayesian particle polishing and CTF refinement were used to generate shiny particles and optimize per-particle defocus parameters[75]. The final cryo-EM maps were B-factor sharpened with RELION's post-processing function applying a global, soft-edged volume mask low-passed filtered to 15 Å. Resolution values were calculated using gold-standard Fourier Shell Correlation (FSC) in RELION[76].

**Model building and refinement**. Model building was initiated by docking the structure of ScORC•DNA (PDB 5zr1[6]) into the cryo-EM maps of ScORC•DNA and ScORC•DNA•Cdc6, followed by manual rebuilding in COOT[77]. In contrast to previous low-resolution models of ScCdc6 and ScOrc6-CB$_N$[8,9,34,57], the high resolution of our cryo-EM maps enabled manual de novo building of Cdc6 and of the N-terminal cyclin box fold of Orc6 and clear identification of many side chain densities and unambiguous amino acid register assignment. The model coordinates were improved by iterative real-space refinement in PHENIX[78,79] (with secondary structure restraints for protein and DNA, and rotamer and Ramachandran restraints for protein) and manual rebuilding in COOT. Unsharpened, sharpened, and PHENIX density-modified cryo-EM maps[80] were all used for map interpretation, model building, and refinement and were smoothened by resampling in COOT by a factor of 1.5 to aid interpretation. The density-modified maps allowed clear resolution of well-ordered water molecules, which were placed in the cryo-EM maps using a combination of PHENIX.DOUSE and the "find water" option in COOT, followed by manual inspection. The final models were validated using MolProbity[81] (Supplementary Table 1) and are in excellent agreement with the cryo-EM maps.

**Structure analysis**. Cryo-EM maps and PDB models were visualized and analyzed using UCSF Chimera[82,83], UCSF ChimeraX[84], COOT[77], and PyMOL (The PyMOL Molecular Graphics System, Version 1.8.2.0 Schrödinger, LLC). PISA[85] as implemented in COOT was used to calculate buried solvent accessible surface areas. UCSF Chimera packages and PyMOL were used to generate figures. Multiple sequence alignments were generated with MAFFT[86,87] and visualized in JALVIEW[88].

**Crosslinking mass spectrometry**. ScORC and ScCdc6 proteins were dialyzed into 25 mM HEPES-KOH (pH 7.6), 250 mM potassium acetate, 10 mM magnesium acetate, 1 mM DTT at 4 °C overnight. Proteins were centrifuged for 10 min at 21000 × g at 4 °C to remove any precipitates that formed during dialysis. The ternary ScORC•DNA•Cdc6 complex was reconstituted as for structural studies but not further purified by gel filtration chromatography. For crosslinking, approximately 35 μg of reconstituted ternary complex was diluted with 73 μL of reconstitution buffer (25 mM HEPES-KOH (pH 7.6), 250 mM potassium acetate, 10 mM magnesium acetate, 1 mM ATP, 1 mM DTT), and CID-cleavable disuccinimidyl sulfoxide (DSSO, spacer arm 10.3 Å, Thermo Fisher Scientific) was added to the protein to a final concentration of 0.5, 2, or 4 mM. After 1 h of incubation at 10 °C, crosslinking reactions were quenched by the addition of Tris-HCl (pH 6.8) to a final concentration of 50 mM and incubated for 1 h at room temperature. The crosslinking reagent was removed by three buffer exchanges into 8 M urea dissolved in 50 mM HEPES-KOH (pH 8.5) using 30 K Amicon Ultra-0.5 mL concentrators (with a wash volume of 400 μL each). Subsequently, the sample was reduced and alkylated for 30 min in 5 mM TCEP and 10 mM 2-chloroacetamide, followed by three buffer exchanges into 8 M urea in 50 mM HEPES-KOH (pH 8.5) as described above. Subsequently, the sample was digested with Lys-C protease (Wako Chemicals, 1:100 enzyme to protein mass ratio) for 4 h at room temperature and with trypsin (Promega, 1:100 enzyme to protein mass ratio) at 37 °C overnight. An additional 4 h incubation followed the next day with freshly added trypsin and 5% acetonitrile. The digest was acidified with trifluoric acid at 1% final concentration, sonicated, and stored at −80 °C.

The digested sample was either injected, loaded, desalted, and then separated on a 50 cm uPAC C18 HPLC column (Pharmafluidics) connected to a modified Digital PicoView nano-source (New Objective), or injected and trapped onto a PepMap 100 C18 2 cm trap using an EASY nLC-1000 system (Thermo Fisher), and then separated on a 15 cm EASY-Spray C18 column (ES801) connected to an EASY-Spray source (all Thermo Fisher Scientific). The following chromatography method was used: 0.1% formic acid (buffer A), 0.1% formic acid in acetonitrile (buffer B), flow rate 500 nl/min (uPAC) or 250 nl/min (EASY), gradient 240 min in total, (mobile phase compositions in % B): 0–5 min 3–7%, 5–195 min 7–22%, 195–225 min 22–80%, 225–240 min 80%. MS spectra were collected on an Orbitrap Fusion Lumos mass spectrometer (Thermo Fisher Scientific) in "MS2_MS3" mode according to Liu et al.[89]. Peptide MS1 precursor ions were measured in the Orbitrap at 120 k resolution with advanced peak determination (APD) feature enabled, and those with assigned charge states between 3 and 8 were subjected to CID–MS2 fragmentation (25% CID collision energy) and fragments detected in the Orbitrap at 30 k resolution. Data-dependent HCD-MS3 scans were performed if a

unique mass difference (Δm) of 31.9721 Da was found in the CID–MS2 scans with detection in the ion trap (35% HCD collision energy).

MS raw data were analyzed in Proteome Discoverer versions 2.4 and 2.5 (Thermo Fisher Scientific) using Sequest[90] search for linear peptides, including crosslinker-modifications, and XlinkX search to identify crosslinked peptides. MS2 fragment ion spectra not indicative of the DSSO crosslink delta mass were searched with the Sequest search engine against a custom protein database containing the expected protein components, a database of proteins previously identified in the Sf9 insect cell expression system[91], as well as a database of contaminants taken from MaxQuant, cRAP, and those commonly identified during inhouse analyses, using the target-decoy strategy[92]. The following variable crosslinker modifications were considered: DSSO Hydrolyzed/+176.014 Da (K); DSSO Tris/+279.078 Da (K), DSSO alkene fragment/+54.011 Da (K); DSSO sulfenic acid fragment/ +103.993 Da (K), as well as Oxidation/+15.995 Da (M) and phosphorylation/ +79.966 Da (S, T, Y). Carbamidomethyl/+57.021 Da (C) was set as a static modification. Trypsin was selected as the cleavage reagent, allowing a maximum of two missed cleavage sites, peptide lengths between 6 and 150, 10 ppm precursor mass tolerance, and 0.02 Da fragment mass tolerance. PSM validation was performed using the Percolator node in PD and a target FDR of 1%. Phosphosites were considered localized above a PTM score of 50 as calculated with the IMP ptmRS node in PD.

XlinkX v2.0 was used to perform a database search against custom protein database containing the expected complex components to identify DSSO-crosslinked peptides and the following variable modification: DSSO Hydrolyzed/ +176.014 Da (K); Oxidation/+15.995 Da (M) and phosphorylation/+79.966 Da (S, T, Y). Crosslink-to-spectrum matches (CSMs) were accepted above an XlinkX score cutoff of 50, requiring a minimum of 3 MS3 spectra for crosslink identification (both peptides must be identified by MS3, at least one peptide as thiol and alkene fragment for DSSO cleavage). MS2-only crosslink identifications were not accepted. The crosslinks from two repeats at three different DSSO concentrations (0.5, 2, and 4 mM) were merged and crosslink network maps generated using the xiNET webserver[93]. For visualization, only crosslinks with a CSM of 4 or greater were included.

**Phosphorylation and dephosphorylation of ScORC**. ORC phosphorylation by CDK was performed in reactions containing 10 μM ScORC and 0.5 μM CDK in phosphorylation buffer (25 mM HEPES (pH 7.6), 300 mM KCl, 10% glycerol, 10 mM MgCl$_2$, 1 mM DTT, and 0.5 mM ATP). ScORC was dephosphorylated at 10 μM using 16.7 units/μL λ phosphatase (New England Biolabs) in buffer containing 25 mM HEPES (pH 7.6), 300 mM KCl, 10% glycerol, 10 mM MgCl$_2$, 1 mM MnCl$_2$, 1 mM DTT, and 0.5 mM ATP. Phosphorylation, dephosphorylation, and untreated control reactions were incubated for 10 min at 30 °C. MnCl$_2$ was added to phosphorylation and control reactions after incubation to adjust its concentration to 1 mM as used in dephosphorylation reactions.

**Mcm7 recruitment assay**. Untreated, CDK phosphorylated, or dephosphorylated MBP-tagged ScORC (with full-length Orc6, Orc6$^{\Delta N120}$, or Orc6$^{\Delta N269}$), 84 bp ARS1 dsDNA, and ScCdc6 were mixed in reaction buffer containing 25 mM HEPES (pH 7.5), 300 mM potassium acetate, 10 mM magnesium acetate, 1 mM DTT, and the slowly hydrolysable nucleotide analog ATPγS. After 10–15 min incubation on ice, ScMcm7 was added. Final reactions contained 0.5 μM ScORC, 0.5 μM DNA, 1 μM ScCdc6, 10 μM ScMcm7, and 2 mM ATPγS. For titrations, 1.5, 5, and 10 μM ScMcm7 were used. 95 μL of assembly reactions were allowed to bind to 50 μL amylose bead slurry (New England Biolabs) for 10 min on ice. Beads were washed three times with low-salt buffer (25 mM HEPES (pH 7.5), 300 mM potassium acetate, 10 mM magnesium acetate, 1 mM DTT). ORC and bound proteins were eluted with low-salt buffer containing 20 mM maltose, separated by SDS-PAGE, and stained with Coomassie. Mcm7 recruitment assays were independently repeated three (for ScMcm7 titrations) to four times (for comparison of different ORC assemblies). ScMcm7 band intensities were quantified with ImageJ, normalized to the λ phosphatase treated samples, and the mean and standard error of the mean of four independent recruitment assays are plotted in Fig. 5c–e.

**Reporting summary**. Further information on research design is available in the Nature Research Reporting Summary linked to this article.

## Data availability

The data that support this study are available from the corresponding author upon reasonable request. The PDB coordinates and cryo-EM maps have been deposited into the Protein Data Bank and Electron Microscopy Data Bank under the following accession numbers: PDB 7TJF and EMD-25924 for ScORC·DNA, PDB 7TJH and EMD-25925 for ScORC·DNA·Cdc6 state I (ODC1), PDB 7TJI and EMD-25926 for ScORC·DNA·Cdc6 state II (ODC2), PDB 7TJJ and EMD-25927 for ScORC·DNA·Cdc6·Orc6-CB$_N$ state I (ODC1·Orc6-CB$_N$), and PDB 7TJK and EMD-25928 for ScORC·DNA·Cdc6·Orc6-CB$_N$ state II (ODC2·Orc6-CB$_N$). The proteomics data have been submitted to the ProteomeXchange Consortium via the PRIDE partner repository with the dataset identifier PXD031103[94]. Source data are provided with this paper.

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

## Acknowledgements

We thank Sandra Muehlhaeusser for assistance with insect cell culturing and Simone Cavadini for help with cryo-EM data collection. This work was supported by the European Research Council under the European Union's Horizon 2020 research and innovation program (ERC-STG-757909 to F.B.), the National Institutes of General Medicine (R01-GM141313 to F.B.), and an Anderson Endowed Postdoctoral Fellowship (to A.K.). O.H. is supported by the NIH predoctoral program in Biophysics (T32-GM008283).

## Author contributions

F.B. conceptualized and supervised the study. R.Y. and F.B. cloned expression constructs and performed biochemical experiments. J.M.S., R.Y., and F.B. purified recombinant proteins. J.M.S. prepared samples for cryo-EM and mass spectrometry, and collected cryo-EM data. F.B. processed cryo-EM data and performed model building and refinement. A.K., R.Y., and O.H. helped with model building and refinement. J.S. performed mass spectrometry. All authors analyzed and interpreted data. J.M.S., R.Y., and F.B. wrote the manuscript with inputs from all other authors.

## Competing interests

The authors declare no competing interests.
