## [Peer Review File · Nature Communications]

A mechanism of origin licensing control through autoinhibition of *S. cerevisiae* ORC·DNA·Cdc6Reviewers' Comments:

Reviewer #1:

Remarks to the Author:

Schmidt and colleagues report the 2.5-2.7Å resolution structures of ORC-Cdc6-DNA. The higher resolution allows for an accurate description of the Orc1>Cdc6 and Orc4>Orc1 ATPase centres, including water molecules expected to be involved in catalysis. The authors also identify a new configuration for ORC6 that folds back and interacts with Orc1 and Cdc6 in a configuration that would prevent helicase loading.

This is a useful study that should be published in Nature communications after addressing the following points:

1. In page 6 the authors compare and contrast the new configurations of ORC-Cdc6 DNA with respect to previously reported ORC-Cdc6-DNA or OCCM structures. The authors do not mention the pre-insertion OCCM intermediate. I believe this structure should be acknowledged.

Page 7 I believe 'transacting' should be 'trans-acting'.

Page 8 in page 8/9 the authors draw a parallel between ORC4 R263 and an Arg coupler element in DnaC. This comparison is condensed in half a sentence, which makes it difficult to digest. I believe this comparison deserves to be expanded on, including a supplementary figure, or else be omitted.

In Figure S4 the authors show the cryoEM map and atomic model of the Orc6 CBN domain. It would be of interest to show a direct comparison with the human and Drosophila CBN.

The finding that the Nterminal domain of Orc6 can bind to Orc1/Cdc6 in a configuration that would impair MCM loading is extremely interesting. Can the author comment of the likelihood that this interaction occurs in cis or in trans? Could N-terminal Orc6 be contributed by a neighbouring ORC particle?

The suggestion that N-terminal Orc6 engagement to Orc1/Cdc6 is promoted through phosphorylation CDK is intriguing. Could the authors clarify whether the phosphorylations identified in Orc6 in the preparation used for cryo-EM indeed included established CDK sites?

Could the authors clarify whether N-terminal Orc6 engaged Orc1/Cdc6 causes any alteration in the ATPase centres of ORC?

Do the authors expect that a truncation of N-terminal ORC6 might cause an accumulation in OCCM intermediates that fail to mature to MO and eventually double hexamers?

Reviewer #2:

Remarks to the Author:

Schmidt et al. report cryoEM structures of yeast ORC (origin recognition complex) complexed with an 84-bp replication origin DNA (termed ScORC-DNA) and Cdc6 (termed ODC). A ScORC-DNA structure of comparable resolution and a similar ODC structure at lower resolution (3.6 Å vs. 2.5-2.7 Å reported in this manuscript) were reported by other authors in 2018 and 2021, respectively. Here with the significantly improved resolution of ODC, the authors show (1) atomic details of the pre-catalytic configuration of the ATPase sites, (2) two different conformations of Cdc6 in DNA binding (ODC1 and ODC2), and (3) a new binding site for the Orc6 N-terminal cyclin-box fold (Orc6-CBN), which inhibits MCM2-7 loading by ODC. The authors verify the newly observed protein interface by crosslinking mass spectrometry. Despite not being the first reported structures, the structural characterizations reported

here are of very high quality and thus solidify and expand our understanding of the yeast ORC-DNA-Cdc6 complex. The manuscript will be improved if the following points can be clarified.

1. In the experimental description and final ODC structures, the authors seem to suggest that there is no ATP hydrolysis in the ODC complex. Can the authors provide ATPase analysis in solution and measure the basal level of ODC ATPase activity at room temperature and 30°C?
2. In Fig. S1, the authors suggest that Orc6 is heterogeneously phosphorylated. Presumably, the structural characterization reported here represents the heterogeneous Orc6. Is it possible to determine the homogeneous ODC structure after dephosphorylation by λ phosphatase treatment?
3. Fig. S2 shows that Orc6-CBN is either flexible (invisible) or docked between Orc1 and Cdc6 (resolved). Is the docked Orc6-CBN, which inhibits MCM loading, correlated with the unphosphorylated state of Orc6? Could the authors use cryoEM and λ phosphatase treatment to determine whether the two Orc6-CBN conformations are correlated with the phosphorylation states of Orc6?
4. The two conformations of Cdc6 (ODC1 and ODC2) are not shown in the summary mechanistic proposal of MCM loading in Fig. 5.
5. In Fig. 5, is the partially attached OCCM real (experimentally observed) or a hypothesis?
6. It will be helpful to include small cartoon diagrams (similar to those shown in Fig. 5) to depict and simplify the complicated structures shown in Figs. 1A, 1B, and 3A. Additional labels of AAA+, WB and Lid domains would also be helpful.
7. Cdc6 is shown in four different colors (red, blue, yellow and cyan) in Fig. 1, and may cause confusion with similarly colored Orc3 and Orc5. The color of Cdc6 in the boxed region in Fig. 1A doesn't match Fig. 1E and 1F.

Reviewer #3:

Remarks to the Author:

The manuscript by Bleichert and colleagues describes structural studies of early intermediates in budding yeast helicase loading. In particular, they describe high resolution structures of ORC bound to DNA and ORC and Cdc6 bound to DNA. The structures were obtained in the presence of ATP and an 84 bp DNA molecule without using a cross-linking agent. This is in contrast to previous studies that used a non-hydrolysable analog of ATP and cross-linking agent. The structures show multiple conformations of the ORC-Cdc6-DNA complex, providing new insight into the dynamics of DNA binding by the helicase loading protein and a potentially new regulatory mechanism for loading of the Mcm2-7 helicase on DNA.

The high resolution (sub-3 angstroms) structures reveal several previous new aspects of ORC and Cdc6 structures. The authors see two new forms of the ORC-Cdc6-DNA (ODC) structure and in one there is a previously undetected interaction of Cdc6 with the DNA. This is similar to previous observations that the same lab has made with the *Drosophila* ORC-Cdc6-DNA structure. Because only a subset of the structures have this interaction (ODC2 but not ODC1) it seems that this interaction is not always in place. Because it is near the ATP binding region and involves the AAA+ B-motif, it is tempting to suggest that ATP binding and hydrolysis could be involved although the authors do not discuss any difference in the bound nucleotide in these two structures. The authors also suggest that there is a progression between the ODC1 to ODC2 structures but the current data does not support a kinetic order to these two structures and this should be acknowledged.

Much of the paper discusses the two ATPase sites of in the ORC-DNA and ORC-Cdc6-DNA (ODC) structure. For both the Orc1-Orc4 and Cdc6-Orc1 ATPase sites the authors observe that the water molecules required for hydrolysis are not correctly positioned. This observation explains why DNA bound ORC is unable to hydrolyze ATP. The same conformation is observed in the OD and the ODC structures suggesting that ORC ATP hydrolysis does not occur in the context of the ODC either (note that the time of ORC-ATP hydrolysis is unknown during helicase loading and is not required for this event).

The most surprising finding in the paper comes from an additional ORC-Cdc6-DNA structure. In this structure, the N-terminal domain of Orc6, previously found to interact with Mcm2/Mcm5 interface, interacts with Orc1, 123Å away from the C-terminal domain. This interaction is further supported by crosslinking mass spectrometry data. The Orc6 N-terminal domain occupies the space where the Mcm7 Winged Helix would be in the ORC-Cdc6-Cdt1-Mcm2-7 (OCCM) complex (the next stable intermediate in the helicase loading event), suggesting that this intermediate could block OCCM formation. The authors also show data that in a recent "semi-attached" OCCM structure, the density for the Orc6 N-terminal domain was improperly attributed to the Mcm7 Winged-Helix domain. Although this is an intriguing observation, the authors provide no evidence that this structure has biological relevance. For example, it is possible that this structure only forms when Orc6 is phosphorylated, a modification known to inhibit helicase loading (but not initial Mcm2-7 recruitment). This is possible since the authors are using a preparation in which Orc6 is phosphorylated. An argument that this is not a phosphorylation-dependent autoinhibited complex can be raised from the "semi-attached" structure. In this case having Orc6 bound at the same site does not seem to prevent subsequent OCCM formation. Indeed, there is nothing to say that binding to this site acts as a chaperone to facilitate Mcm2-7's subsequent binding (i.e. it is an interaction is auto-stimulatory rather than inhibitory).

In summary, the new observations made with these higher resolution structures are for the most part minor revisions of our view of ORC and Cdc6 interactions with DNA and ATP. Although it is nice to know the specific change in the ATPase sites that prevents hydrolysis and that there is an additional DNA contact, in neither case do these observations significantly change our view of the events of helicase loading (e.g. they do not give insights into the role or the timing of ATP binding and hydrolysis). The most impactful part of the paper is the identification of a new binding site for Orc6 in the ODC structure. If the authors can provide some evidence that these interactions are important for helicase loading or it's inhibition by CDK phosphorylation then this paper would be well suited for publication. For example, do mutants in this site reduce CDK-dependent inhibition of helicase loading (e.g. by showing that mutations that would prevent this interaction facilitate helicase loading). Or, does CDK phosphorylation inhibit this interaction (e.g. as measured by MS-XLinking). Without such data it is premature to state that this structure represents an auto-inhibited state.

We thank the reviewers for the careful review of our manuscript and their insightful comments. We are grateful for the reviewers' enthusiasm for our work, especially for our discovery of a new conformation of the yeast ORC-DNA-Cdc6 complex, which results from docking of the Orc6 N-terminal cyclin-box domain (Orc6-CB_N) onto the ORC-Cdc6 ring and which we hypothesized to hinder Mcm2-7 loading. However, reviewer 3 indicated that additional evidence is needed to support our suggested mechanism of origin licensing control through autoinhibition of ORC-DNA-Cdc6.

In the enclosed revision, we have addressed this concern and added new biochemical experiments showing that Mcm7 recruitment to ORC-DNA-Cdc6 is regulated by Orc6-CB_N in response to ORC phosphorylation. These new findings are included in **Figure 5** and **Supplemental Figure S12** of the revised manuscript. A point-by-point response to all other issues raised by the referees and a detailed description of changes are included in blue below.

Revised figures:

- Figure 1
- Figure 3
- Figure 5 (new)
- Figure 6
- Supplemental Figure S4
- Supplemental Figure S5 (new)
- Supplemental Figure S11 (new data added)
- Supplemental Figure S12 (new)

Reviewer #1:

Schmidt and colleagues report the 2.5-2.7Å resolution structures of ORC-Cdc6-DNA. The higher resolution allows for an accurate description of the Orc1>Cdc6 and Orc4>Orc1 ATPase centres, including water molecules expected to be involved in catalysis. The authors also identify a new configuration for ORC6 that folds back and interacts with Orc1 and Cdc6 in a configuration that would prevent helicase loading.

This is a useful study that should be published in Nature communications after addressing the following points.

We thank the reviewer for the positive assessment of our work and for supporting publication in Nature Communications.

1. In page 6 the authors compare and contrast the new configurations of ORC-Cdc6 DNA with respect to previously reported ORC-Cdc6-DNA or OCCM structures. The authors do not mention the pre-insertion OCCM intermediate. I believe this structure should be acknowledged.

We thank the reviewer for this comment. We have indeed attempted to compare the ring conformations of our ODC structures to that of the pre-insertion OCCM in addition to the loaded OCCM shown in **Figure 1D**. However, the relatively low resolution (8.1 Å) of the cryo-EM map of the pre-insertion OCCM precludes an accurate placement of the AAA+ domains into the cryo-EM map that is needed to warrant such a comparison. To avoid overinterpretation of the available structural data for the pre-insertion OCCM, we prefer not to include this intermediate in

Figure 1D. We note that we have referenced the pre-insertion OCCM in revised **Figure 6** (former **Figure 5**) and on page 16 in the main text.

2. Page 7 I believe 'transacting' should be 'trans-acting'.

We thank the referee for spotting this typo, which we have corrected.

3. Page 8 in page 8/9 the authors draw a parallel between ORC4 R263 and an Arg coupler element in DnaC. This comparison is condensed in half a sentence, which makes it difficult to digest. I believe this comparison deserves to be expanded on, including a supplementary figure, or else be omitted.

We appreciate the reviewer's suggestion and have expanded on the comparison of Orc4-R263 and the DnaC Arg-coupler in the main text on pages 8-9 of the revised manuscript and in new **Supplemental Figures S5**.

4. In Figure S4 the authors show the cryoEM map and atomic model of the Orc6 CBN domain. It would be of interest to show a direct comparison with the human and *Drosophila* CBN.

As suggested, we have now added a comparison of the budding yeast and human Orc6-CB_Ns in **Supplemental Figure S4** (in new panel C) and included a comment in the main text on page 13. We could not show the structure of the *Drosophila* Orc6-CB_N because it has not yet been determined.

5. The finding that the Nterminal domain of Orc6 can bind to Orc1/Cdc6 in a configuration that would impair MCM loading is extremely interesting. Can the author comment on the likelihood that this interaction occurs in cis or in trans? Could N-terminal Orc6 be contributed by a neighbouring ORC particle?

The reviewer raises an excellent point. Currently, we do not know whether the interaction between Orc6-CB_N and the ORC-Cdc6 ring occurs *in cis* and/or *in trans*. Our current structural knowledge is consistent with both possibilities. We have modified the main text on pages 16-17 of the revised manuscript to clarify this point.

6. The suggestion that N-terminal Orc6 engagement to Orc1/Cdc6 is promoted through phosphorylation CDK is intriguing. Could the authors clarify whether the phosphorylations identified in Orc6 in the preparation used for cryo-EM indeed included established CDK sites?

Using mass spectrometry, we have identified numerous phosphorylation sites in Orc6, including two of the established CDK sites, S106 and S116, which are located near the CB_N. Unfortunately, we observed no peptides for the region surrounding S123, hence we were not able to assess the phosphorylation status of this residue. We have now marked these experimentally determined phosphosites in our ORC preparation in the alignment in **Supplemental Figure S11** and have updated the proteomics data submission to the PRIDE repository accordingly.

Related to this question, please also see our answers to points #6 and #8 of referee 3, which explain follow-up experiments showing that CDK phosphorylation of ORC prevents Mcm7 recruitment to ORC-Cdc6, consistent with a phosphorylation-dependent docking of Orc6-CB_N to the ORC-Cdc6 ring.

7. Could the authors clarify whether N-terminal Orc6 engaged Orc1/Cdc6 causes any alteration in the ATPase centres of ORC?

We observe no obvious changes in the ATPase centers in our structures upon docking of the N-terminal Orc6 domain onto Orc1/Cdc6. We have now clarified this point on pages 11-12 of the revised manuscript.

8. Do the authors expect that a truncation of N-terminal ORC6 might cause an accumulation in OCCM intermediates that fail to mature to MO and eventually double hexamers?

It has previously been shown by the Costa laboratory that deletion of N-terminal Orc6 residues 1-119 leads to OCCM intermediates that fail to mature to MO and double hexamers (Miller *et al.*, Nature 2019). Specifically, the efficiency of MO and double hexamer formation decreased by 2- and 2.6-fold, respectively. Very recently, similar results have been obtained using single molecule FRET approaches by the Bell lab (Gupta *et al.*, eLife 2021). We have updated the main text on page 13 to reference these studies.

Reviewer #2:

Schmidt et al. report cryoEM structures of yeast ORC (origin recognition complex) complexed with an 84-bp replication origin DNA (termed ScORC-DNA) and Cdc6 (termed ODC). A ScORC-DNA structure of comparable resolution and a similar ODC structure at lower resolution (3.6 Å vs. 2.5-2.7 Å reported in this manuscript) were reported by other authors in 2018 and 2021, respectively. Here with the significantly improved resolution of ODC, the authors show (1) atomic details of the pre-catalytic configuration of the ATPase sites, (2) two different conformations of Cdc6 in DNA binding (ODC1 and ODC2), and (3) a new binding site for the Orc6 N-terminal cyclin-box fold (Orc6-CBN), which inhibits MCM2-7 loading by ODC. The authors verify the newly observed protein interface by crosslinking mass spectrometry. Despite not being the first reported structures, the structural characterizations reported here are of very high quality and thus solidify and expand our understanding of the yeast ORC-DNA-Cdc6 complex. The manuscript will be improved if the following points can be clarified.

We are thankful that the reviewer appreciates the high-quality of our work and the new insights it reveals for understanding the function of ORC-DNA-Cdc6.

1. In the experimental description and final ODC structures, the authors seem to suggest that there is no ATP hydrolysis in the ODC complex. Can the authors provide ATPase analysis in solution and measure the basal level of ODC ATPase activity at room temperature and 30°C?

We did not intend to suggest that there is no ATP hydrolysis in the ODC complex, and we apologize for this misunderstanding. ATP hydrolysis in ORC is inhibited by DNA binding (Klemm *et al.*, Cell 1997), and ODC's ATPase activity is kept low and stimulated by Mcm3 or MCM

binding (Frigola *et al.*, Nature 2013). The low ATPase activity can be explained by the configuration of the ATPase sites in our structures. We have modified the text on pages 7 and 10 to better clarify this point. In terms of measuring the ATPase activity of ODC at room temperature and 30°C, we are unfortunately currently limited by the amount of ORC needed to perform spectrophotometric ATPase assays set up in our laboratory.

2. In Fig. S1, the authors suggest that Orc6 is heterogeneously phosphorylated. Presumably, the structural characterization reported here represents the heterogeneous Orc6. Is it possible to determine the homogeneous ODC structure after dephosphorylation by λ phosphatase treatment?

The reviewer is correct that our cryo-EM dataset contains a mixture of phosphorylated and unphosphorylated Orc6, as can be seen in **Supplemental Figure S1B**. Likewise, our cryo-EM dataset contains a mixture of ODC complexes, both with flexible Orc6-CB_N and with Orc6-CB_N docked onto the ORC-Cdc6 ring (**Supplemental Figure S2**). While we think that it is possible to homogenize the ODC structure with respect to Orc6-CB_N docking by dephosphorylation with λ phosphatase, we do not anticipate observing additional structural states to those in our heterogeneously phosphorylated ODC sample. Indeed, one of the strengths of cryo-EM is to allow structure determination of different states in heterogeneous samples. However, we have homogenized the phosphorylation states of our ORC preparations by λ phosphatase and CDK treatment for biochemical studies included in **new Figure 5**, which show that CDK phosphorylation of ORC prevents Mcm7 binding to ORC-Cdc6 in an Orc6-CB_N-dependent manner, while λ phosphatase treatment permits Mcm7 recruitment. These new results are consistent with competition between Orc6-CB_N and Mcm7-WH domain docking to ORC-Cdc6 as predicted by our structural data. We also refer the reviewer to our answers to points #6 and #8 of referee 3.

3. Fig. S2 shows that Orc6-CBN is either flexible (invisible) or docked between Orc1 and Cdc6 (resolved). Is the docked Orc6-CBN, which inhibits MCM loading, correlated with the unphosphorylated state of Orc6? Could the authors use cryoEM and λ phosphatase treatment to determine whether the two Orc6-CBN conformations are correlated with the phosphorylation states of Orc6?

We have attempted to correlate the phosphorylation state of ORC with Orc6-CB_N docking using negative stain EM after inserting localization tags such as GFP into Orc6. Unfortunately, we were not able to see density corresponding to GFP in 2D class averages due to residual flexibility of the tag. As an alternative approach, we have devised a biochemical assay using Mcm7 binding as readout of Orc6-CB_N docking onto the ORC-Cdc6 ring, since both binding events are predicted to be mutually exclusive based on existing structural data (this manuscript and Yuan *et al.*, NSMB 2017). As mentioned in the response to the previous point, phosphorylation by CDK decreases Mcm7 recruitment, while deletion of the Orc6-CB_N abrogates CDK-mediated inhibition of Mcm7 binding. These data are included in **new Figure 5** and support our model of phosphorylation-dependent regulation of Orc6-CB_N binding to ORC-Cdc6.

4. The two conformations of Cdc6 (ODC1 and ODC2) are not shown in the summary mechanistic proposal of MCM loading in Fig. 5.

We apologize for this oversight. We have added both states to the summary figure (**now revised Figure 6**).

5. In Fig. 5, is the partially attached OCCM real (experimentally observed) or a hypothesis?

The partially attached OCCM (or semi-attached OCCM) that was schematized in **Figure 5 (now revised Figure 6)** is a real intermediate that has been visualized previously in 2D cryo-EM class averages by Yuan *et al.* (PNAS 2020) and partially in 3D, with the caveat that the map region previously assigned to be the WH domain of Mcm7 is actually Orc6-CB_N. The cryo-EM map of the semi-attached OCCM is shown in **Supplemental Figure S10**. Note that apart from the Mcm3-WH domain, Mcm2-7 is not structurally resolved in the 3D map because of flexibility. Since the semi-attached OCCM had been defined to contain attached Mcm3-WH and Mcm7-WH domains, we prefer to refer to OCCM in which only the Mcm3-WH domain is docked as partially attached OCCM to avoid confusion.

6. It will be helpful to include small cartoon diagrams (similar to those shown in Fig. 5) to depict and simplify the complicated structures shown in Figs. 1A, 1B, and 3A. Additional labels of AAA+, WB and Lid domains would also be helpful.

We thank the reviewer for this suggestion. We have now included cartoon diagrams in **Figures 1A, 1B, and 3A**, and also labeled Cdc6 domains in **Figure 1C**.

7. Cdc6 is shown in four different colors (red, blue, yellow and cyan) in Fig. 1, and may cause confusion with similarly colored Orc3 and Orc5. The color of Cdc6 in the boxed region in Fig. 1A doesn't match Fig. 1E and 1F.

In **Figure 1C and 1G**, the blue and yellow colors represent the two different ODC states (ODC1 and ODC2) that can be seen in **Supplemental Figure S2** rather than Cdc6. Throughout the manuscript, we consistently color Cdc6 red unless we are differentiating between ODC1 and ODC2 states. The cyan color in **Figures 1E-F** is used to highlight the B-loop region of Cdc6 and its contacts with DNA, while the remainder of Cdc6 remains colored red in these panels. To avoid confusion, we have modified the legend of **Figure 1**.

Reviewer #3:

1. The manuscript by Bleichert and colleagues describes structural studies of early intermediates in budding yeast helicase loading. In particular, they describe high resolution structures of ORC bound to DNA and ORC and Cdc6 bound to DNA. The structures were obtained in the presence of ATP and an 84 bp DNA molecule without using a cross-linking agent. This is in contrast to previous studies that used a non-hydrolysable analog of ATP and cross-linking agent. The structures show multiple conformations of the ORC-Cdc6-DNA complex, providing new insight into the dynamics of DNA binding by the helicase loading protein and a potentially new regulatory mechanism for loading of the Mcm2-7 helicase on DNA.

We are delighted to hear that the reviewer agrees that our structural work provides new insights into the dynamics and regulation of the Mcm2-7 helicase loader.

2. The high resolution (sub-3 angstroms) structures reveal several previous new aspects of ORC and Cdc6 structures. The authors see two new forms of the ORC-Cdc6-DNA (ODC) structure and in one there is a previously undetected interaction of Cdc6 with the DNA. This is similar to previous observations that the same lab has made with the *Drosophila* ORC-Cdc6-DNA structure. Because only a subset of the structures have this interaction (ODC2 but not ODC1) it seems that this interaction is not always in place. Because it is near the ATP binding region and involves the AAA+ B-motif, it is tempting to suggest that ATP binding and hydrolysis could be involved although the authors do not discuss any difference in the bound nucleotide in these two structures.

We apologize that this comparison was not clear in the original manuscript. We have included a detailed comparison of the ODC1 and ODC2 ATPase sites in **Supplemental Figure S6** (**Supplemental Figure S5** in the original submission). We have now also modified the main text in the revised manuscript and discuss on page 9 that the ATPase sites of ODC1 and ODC2 are configured very similarly to each other.

3. The authors also suggest that there is a progression between the ODC1 to ODC2 structures but the current data does not support a kinetic order to these two structures and this should be acknowledged.

We thank the reviewer for raising this point. We agree that the cryo-EM structures are static and do not provide kinetic information. We have revised the main text on page 6 to avoid inferences of kinetic order.

4. Much of the paper discusses the two ATPase sites of in the ORC-DNA and ORC-Cdc6-DNA (ODC) structure. For both the Orc1-Orc4 and Cdc6-Orc1 ATPase sites the authors observe that the water molecules required for hydrolysis are not correctly positioned. This observation explains why DNA bound ORC is unable to hydrolyze ATP. The same conformation is observed in the OD and the ODC structures suggesting that ORC ATP hydrolysis does not occur in the context of the ODC either (note that the time of ORC-ATP hydrolysis is unknown during helicase loading and is not required for this event).

We appreciate the reviewer's detailed summary of our work. We would like to point out that ORC-DNA-Cdc6 has a low (but non-zero) basal level of ATPase activity, which is further stimulated after MCM recruitment and specifically by Mcm3 binding (Frigola *et al.*, Nature 2013). We have now clarified this point on page 7 of the revised manuscript. We envision that Mcm3 binding allows ORC-DNA-Cdc6 to more frequently sample a hydrolysis-competent state, resulting in increased ATPase rates.

5. The most surprising finding in the paper comes from an additional ORC-Cdc6-DNA structure. In this structure, the N-terminal domain of Orc6, previously found to interact with Mcm2/Mcm5 interface, interacts with Orc1, 123Å away from the C-terminal domain. This interaction is further supported by crosslinking mass spectrometry data. The Orc6 N-terminal domain occupies the space where the Mcm7 Winged Helix would be in the ORC-Cdc6-Cdt1-Mcm2-7 (OCCM) complex (the next stable intermediate in the helicase loading event), suggesting that this intermediate could block OCCM formation.

We appreciate that the reviewer considers the new ODC state, in which the N-terminal Orc6 domain is docked onto the ORC-Cdc6 ring, interesting and surprising.

6. The authors also show data that in a recent “semi-attached” OCCM structure, the density for the Orc6 N-terminal domain was improperly attributed to the Mcm7 Winged-Helix domain. Although this is an intriguing observation, the authors provide no evidence that this structure has biological relevance. For example, it is possible that this structure only forms when Orc6 is phosphorylated, a modification known to inhibit helicase loading (but not initial Mcm2-7 recruitment). This is possible since the authors are using a preparation in which Orc6 is phosphorylated. An argument that this is not a phosphorylation-dependent autoinhibited complex can be raised from the “semi-attached” structure. In this case having Orc6 bound at the same site does not seem to prevent subsequent OCCM formation. Indeed, there is nothing to say that binding to this site acts as a chaperone to facilitate Mcm2-7's subsequent binding (i.e. it is an interaction is auto-stimulatory rather than inhibitory).

We respectfully disagree with the reviewer's viewpoint that the previously determined “semi-attached” OCCM structure (Yuan *et al.*, PNAS 2020) argues that Orc6-CB_N docking does not prevent formation of the fully attached OCCM, or that the N-terminal Orc6 domain is auto-stimulatory to MCM recruitment for the following reasons:

First, Yuan *et al.* (PNAS 2020) provide no experimental evidence that the semi-attached OCCM can mature to the fully attached OCCM. The structures presented in Yuan *et al.* are derived from distinct particles, which may contain differentially phosphorylated ORC. Maturation of only the unphosphorylated ODC to the fully attached OCCM, as we suggest, would yield the same data as reported by Yuan *et al.*, which are thus fully consistent with our proposed model.

Second, an autostimulatory role of the Orc6-CB_N for MCM recruitment predicts that deletion of this domain reduces MCM recruitment. However, the Costa laboratory has previously shown that ORC containing a deletion of the N-terminal domain of Orc6 recruits MCM as efficiently as full-length ORC (Miller *et al.*, Nature 2019). Likewise, Gupta *et al.* recently reported that the same deletion in Orc6 supported recruitment of a single MCM hexamer in single molecule studies (Gupta *et al.*, eLife 2021). These data are inconsistent with an autostimulatory role of the Orc6-CB_N during MCM recruitment.

Third, we have performed additional biochemical experiments that show a) phosphorylation of ORC by CDK reduces the recruitment of Mcm7 to ORC-DNA-Cdc6 while dephosphorylation of ORC increases Mcm7 recruitment, and b) CDK-mediated inhibition of Mcm7 recruitment is dependent on the N-terminal domain of Orc6 and can be overcome by a deletion of Orc6-CB_N or Orc6-CB_N+linker. These new biochemical data are fully consistent with our proposed model that Orc6 phosphorylation promotes Orc6-CB_N docking onto the ORC-Cdc6 ring, thereby blocking the binding site for the Mcm7-WH domain during MCM recruitment. These findings are now included as **new Figure 5** and **Supplemental Figure 12** and support the functional relevance of our structural findings.

7. In summary, the new observations made with these higher resolution structures are for the most part minor revisions of our view of ORC and Cdc6 interactions with DNA and ATP. Although it is nice to know the specific change in the ATPase sites that prevents hydrolysis and that there is an additional DNA contact, in neither case do these observations significantly change our view of the events of helicase loading (e.g. they do not give insights into the role or

the timing of ATP binding and hydrolysis).

We respectfully disagree with the reviewer's viewpoint that our structures are only minor revisions of previous findings. The improved resolution of our structures allows us a) to identify new, functionally relevant conformational states of ORC·DNA·Cdc6 (see also our answers to points #6 and #8), b) to correct register shifts and provide more accurate pdb models, c) to assign side chain positions and rotamer states more precisely, and d) to locate essential water molecules in the active sites of ORC and ORC·Cdc6 for the first time. These structural insights establish that the ATPase sites predominantly reside in a pre-hydrolysis state (which was not visible in previous structures because of lower resolution), and resolve long-standing questions of how the ATPase activity is repressed in ORC-containing assemblies prior to MCM recruitment.

8. The most impactful part of the paper is the identification of a new binding site for Orc6 in the ODC structure. If the authors can provide some evidence that these interactions are important for helicase loading or its inhibition by CDK phosphorylation then this paper would be well suited for publication. For example, do mutants in this site reduce CDK-dependent inhibition of helicase loading (e.g. by showing that mutations that would prevent this interaction facilitate helicase loading). Or, does CDK phosphorylation inhibit this interaction (e.g. as measured by MS-XLinking). Without such data it is premature to state that this structure represents an auto-inhibited state.

We thank the reviewer for these suggestions. As described in our response to point #6, we have added new biochemical results in **Figure 5** and **Supplemental Figure S12** that show phosphorylation-dependent regulation of Mcm7 recruitment that relies on the presence of the Orc6-CB_N. Unlike full-length Orc6, deletion mutants lacking the N-terminal Orc6 cyclin box fold in isolation or in combination with the linker region fail to inhibit Mcm7 recruitment to ORC·DNA·Cdc6 in response to CDK phosphorylation as predicted by our model. We note that we were not able to examine MCM loading because the Orc6-CB_N facilitates MO complex and MCM double hexamer formation (Miller *et al.*, Nature 2019; Gupta *et al.*, eLife 2021), using the same Orc6-CB_N surface region for binding to Mcm2/6 in MO and to ORC·Cdc6 (see **Figure 4**). Generation of separation-of-function mutants is thus extremely challenging if not impossible. However, we hope that our new findings on Mcm7 recruitment and its regulation by CDK phosphorylation sufficiently address the concern of this reviewer.

References:

Miller, T. C. R., Locke, J., Greiwe, J. F., Diffley, J. F. X. & Costa, A. Mechanism of head-to-head MCM double-hexamer formation revealed by cryo-EM. Nature 575, 704-710 (2019).

Gupta, S., Friedman, L. J., Gelles, J. & Bell, S. P. A helicase-tethered ORC flip enables bidirectional helicase loading. eLife 10, e74282 (2021).

Klemm, R. D., Austin, R. J. & Bell, S. P. Coordinate binding of ATP and origin DNA regulates the ATPase activity of the origin recognition complex. Cell 88, 493-502 (1997).

Frigola, J., Remus, D., Mehanna, A. & Diffley, J. F. ATPase-dependent quality control of DNA replication origin licensing. Nature 495, 339-343 (2013).

Yuan, Z. et al. Structural basis of Mcm2-7 replicative helicase loading by ORC-Cdc6 and Cdt1. Nat Struct Mol Biol 24, 316-324 (2017).

Yuan, Z. et al. Structural mechanism of helicase loading onto replication origin DNA by ORC-Cdc6. *Proc Natl Acad Sci U S A* 117, 11747-11756 (2020).

Reviewers' Comments:

Reviewer #1:

Remarks to the Author:

The authors have addressed my concerns in full.

In addition, the new data on CDK modulation of Mcm7 recruitment (Figure 5) significantly to the impact of the paper. In my opinion this works is now ready for publication in Nature Communications.

Reviewer #2:

Remarks to the Author:

The authors have addressed my concerns.

Reviewer #3:

Remarks to the Author:

The revised manuscript by Bleichert and colleagues is significantly improved. In addition to presenting a higher resolution version of the ORC-Cdc6-DNA structure than previously published, the revised manuscript has added new data showing that ORC phosphorylation prevents association of Mcm7 with ORC. When coupled with the intriguing observation of a new interaction between the Orc6-Nterminal domain with the region of ORC-Cdc6-DNA that Mcm7 would normally bind in the context of the initial OCCM intermediate, these observations suggest important new models for how ORC phosphorylation inhibits helicase loading. The addition of this data combined with the previous studies make the manuscript of significant interest to the cell cycle and DNA replication fields and appropriate for publication.